# Improving Raw Readings from Low-Cost Ozone Sensors Using Artificial Intelligence for Air Quality Monitoring

Guillem Montalban-Faet, Eric Meneses-Albala, Santiago Felici-Castell, Juan J. Perez-Solano, and Jaume Segura-Garcia

Departament de Informàtica, ETSE, Universitat de València, Avd. de la Universidad S/N, 46100 Burjassot, (Valencia), Spain

**Correspondence:** Santiago Felici-Castell (felici@uv.es)

**Abstract.** Ground-level ozone ($O_3$) is a highly oxidizing gas with very reactive properties, harmful at high levels, and generated by complex photochemical reactions when primary pollutants from the combustion of fossil materials react with sunlight. Thus, its concentration indicates the activity of other air pollutants and plays a crucial role in smart cities. With the growing interest in high-resolution Air Quality (AQ) monitoring, low-cost ozone sensors present an interesting alternative, although they

lack accuracy and suffer from cross-sensitivity issues. In this context, artificial intelligence techniques, particularly ensemble Machine Learning (ML) models, can improve the raw readings from these sensors by incorporating additional environmental information to minimize inaccuracies and nonlinearities, as well as by including metadata to account for sensor aging effects and improve the models based on road traffic patterns. In this paper, based on the low-cost ZPHS01B multisensor module with nine sensors, we analyze, propose, and compare different techniques using four ML models in a low $O_3$ concentration

scenario (mean value of 55.72 $\mu g/m^3$). We carried out a thorough exploratory data analysis process to extract the main features (variables) and performed hyperparameter optimization for the different models. As a result, we reduced the estimation error by approximately 94.05%. In particular, using the Gradient Boosting algorithm, we achieved a Mean Absolute Error (MAE) of 4.022 $\mu g/m^3$ and a Mean Relative Error (MRE) of 7.21%, outperforming related work while using a module approximately ten times less expensive. To carry out this work, we generated two datasets in the city of Valencia (Spain), at two different

locations with the same characteristics (close to the ring road but separated by 4.1 km), with 165 and 239 days.

## 1 Introduction

Air Quality (AQ) is a fundamental aspect of environmental health, addressing the composition and purity of atmospheric gases, in terms of fine Particulate Matter (PM), Nitrogen Oxides (such as $NO$, $NO_2$ and total $NO_x$), Sulfur Dioxide ($SO_2$), Total Volatile Organic Compounds ($TVOC$) and ground-level Ozone ($O_3$), now on named simply as $O_3$.

AQ has a direct impact on both human health and the environment (Manisalidis et al. (2020)). According to World Health Organization (WHO) (H. Adair-Rohani (2024)), 99% of the world's population breathes air that exceeds the limit values of the recommended safety Air Quality Guideline (AQG) (Organization et al. (2021)). This guideline specifies recommended levels for these pollutants for both short-term and long-term exposure. It is regularly reviewed and updated to incorporate the

latest scientific evidence on the health effects of air pollution. This helps governments and authorities establish and implement policies to protect human health from the adverse effects of air pollution.

Among these pollutants, we focus on $O_3$, a highly oxidizing gaseous pollutant, that has very reactive properties and is harmful at high levels. Notice that in this AQG with regard to $O_3$, the target is to achieve a concentration of 100 $\mu g/m^3$ measured on average of daily maximum 8 hours. Continued exposure to levels above those recommended by this AQG may lead to respiratory irritation, lung inflammation, aggravation of respiratory diseases such as asthma or bronchitis, cell damage and may have associated effects on the cardiovascular system. Those at the highest risk include children, older adults, people with respiratory or heart conditions, and individuals who spend significant time outdoors (Garcia et al. (2021)).

This gas is very important to monitor, because it is called a secondary pollutant, which is generated in cities by complex photochemical reactions when primary pollutants from combustion of fossil materials (such as $NO$, $NO_2$ and $SO_2$) react with sunlight (Seinfeld and Pandis (2016)). Thus, its concentration indicates the activity of other air pollutants and plays a crucial role in AQ monitoring systems in smart cities to help their citizens improve their quality of life. It is worth mentioning that it is being recommended to increase the spatial sampling resolution of this pollutant, ideally at least one sample per 100 $m^2$, according to Annex III-B of the European (Directive 2008/50/EC (2008)). And Low-Cost Sensor (LCS) are becoming increasingly important, an interesting alternative, but they do not have good accuracy (Borrego et al. (2016)) in comparison with the regulated equipment, due to limitations in their sensing technology, lack of frequent calibration, sensitivity to environmental factors, cross-sensitive issues, use of less durable materials and the absence of rigorous certification processes. While regulated equipment uses advanced technologies and is subject to strict standards of accuracy and reliability, LCS are designed to offer basic monitoring at a low price, which involves sacrifices in accuracy and durability. So, in this context it is a challenge to estimate the regulated measurements from these LCS with a reduced error (García et al. (2022); Borrego et al. (2016)).

Artificial Intelligence (AI) techniques are valuable for environmental research due to their capacity to process large datasets and identify patterns that enhance system explainability and clarify the behavior of these AQ parameters (Zhu et al. (2023)).

In this paper, we show that Machine Learning (ML) models, particularly ensemble models, can correct the raw readings from LCS by incorporating additional environmental information, such as Temperature (Temp), Relative Humidity (RH), and other pollutants, as well as by including metadata to account for sensor aging effects and improve the models based on road traffic patterns. With these models, we are able to use these sensors to extend the resolution of AQ monitoring networks at low-cost, but assuming a small error. This is our main objective. We propose and compare different techniques, reducing the estimation error up to 94.05%, in a low $O_3$ concentration scenario (mean value of 55.72 $\mu g/m^3$). In particular, using the Gradient Boosting (GB) algorithm, we achieved a Mean Absolute Error (MAE) of 4.022 $\mu g/m^3$ and a Mean Relative Error (MRE) of 7.21%, outperforming related work, using sensors approximately 10 times less expensive. We also carry out the calibration process using Random Forest (RF), Adaptive Boosting (ADA) and Decision Tree (DT) models. To train and test these models, we use two datasets in the city of Valencia (Spain), at two different locations with the same characteristics (close to the ring road but separated by 4.1 km), with 165 and 239 days.

## 2 Related work

Regarding AQ LCS, due to the increasing market demand, a wide variety of them are available to measure different pollutants, gases and particles. These sensors are available in different price ranges and are more affordable compared to standardized measuring station.

| Module | Sensors | Price range |
|---|---|---|
| SDSO11 (Nova Fitness Co., Ltd. (2024)) | Temp, RH, PM, PA | Low |
| DL-LP8P (DecentLab, Ltd. (2024)) | Temp, RH, $CO_2$, PA | Low |
| MiCS-6814 (SGX, SensorTech (2024)) | $CO, NO_2, C_2H_5OH, NH_3, CH_4$ | Low |
| ZPHS01B (Zhengzhou Winsen Electronics Technology Co. (2024)) | Temp, RH, $PM_{1-10}, CO, CO_2, O_3, NO_2, TVOC$ | Mid-Low |
| Sensit RAMP (Sensit (2024)) | $PM_{2.5}, CO, CO_2, NO, NO_2, O_3$ | High |
| AirSensEUR (Van Poppel et al. (2023)) | $NO, NO_2, O_3, CO, PM_{1-10}, CO_2$ | Mid-High |

**Table 1.** AQ Sensor modules with cost estimate: Low (less than 10$), Mid-Low (100-200$), Mid-High (600-1000$) and High ($\approx$<4000$).

Since in AQ different pollutants are considered and each sensor measures only one, we will analyze sensor modules that embed some of these LCS. A list of these sensor modules with a cost estimate is given in Table 1. The selection criteria of these modules is determined by the related work, selecting those modules which have been considered under a similar studies as the proposed here. We must stress that these modules have different costs due to their quality, order quantity, country, etc. that we can classify in: Low (less than 10$), Mid-Low (100-200$), Mid-High (600-1000$) and High ($\approx$<4000$). A larger selection and comparison of these LCS modules are given in (García et al. (2022)) and (Borrego et al. (2016)).

Note that LCS are designed for basic monitoring at a low-cost, which compromises accuracy and durability. In this list, there are several types of LCS. Optical type sensors, such as SDSO11 (Nova Fitness Co., Ltd. (2024)) and DL-LP8P (DecentLab, Ltd. (2024)), that measure the amount of light absorbed by a given gas. Metal-oxide sensors, such as (SGX, SensorTech (2024)) that measure the change in electrical conductivity on a semiconductor due to the presence of certain gases. Usually this type of sensors are the cheapest and are particularly susceptible to cross sensitivities. And electrochemical sensors that have higher selectivity, good for measuring specific gases, but they are more expensive. Among these, Sensit RAMP (Sensit (2024)) and AirSensEUR (Van Poppel et al. (2023)) use this type of sensors. Finally, the ZPHS01B module (Zhengzhou Winsen Electronics Technology Co. (2024)) integrates optical, metal-oxide and electrochemical sensors and it is a Mid-Low price module with the best *price/sensor* ratio.

Since one of the key points to improve the accuracy of these LCS is the use of marginal information (such Temp, RH as well as other AQ pollutants), exploited using AI techniques (Karagulian et al. (2019); Esposito et al. (2016)) as mentioned before, it is necessary to use multi-gas modules embedding as many AQ LCS as possible.

Thus, among the different low-cost alternatives and taking into account the number of sensors and the *price/sensor* ratio, the ZPHS01B (Zhengzhou Winsen Electronics Technology Co. (2024)) is the AQ sensor module that best meets the needs and objectives of this study at the time of writing, since it embeds 9 different sensors: Temp($^{\circ}$C), RH (%), as well as $CO, CO_2$, $NO_2, O_3$ that are measured in Parts Per Million (ppm), formaldehyde ($CH_2O$) that is measured in mg/$m^3$, PM measured

**Table 2.** AQ information from the ZPHS01B module and units.

| Parameter | [Unit] | Range of Measurement |
|---|---|---|
| Temperature | [ºC] | -20-65 |
| Humidity | [%R.H.] | 0-100 |
| $PM_{2.5}$ | [$\mu g/m^3$] | 0-1000 |
| $TVOC$ | levels | 0-3 |
| $CH_2O$ | [$mg/m^3$] | 0-6.25 |
| $CO_2$ | [ppm] | 0-5000 |
| $CO$ | [ppm] | 0-500 |
| $O_3$ | [ppm] | 0-10 |
| $NO_2$ | [ppm] | 0.1-10 |

in $\mu g/m^3$ and $TVOC$ that is measured using 4 levels according to its concentration (0-very low, 1-low, 2-intermediate and 3-high). Table 2 summarizes all this information. Notice that the $O_3$ sensor used in this module is the electrochemical ZE27-O3 (Corp (2024)) that measures within the range 0-10 ppm with a resolution of 0.01 ppm. It operates with an accuracy of $\pm 0.1$ ppm when the concentration is $\leq 1$ ppm and $\pm 20\%$ when the concentration is above 1 ppm. Also, notice that the PM readings in this module are given for 2.5 (fine particles with a diameter of 2.5 $\mu m$), and $PM_1$ and $PM_{10}$ are estimated from the $CH_{2.5}$ readings.

Based on this ZPHS01B module, there are several research works and projects. In (Coto-Fuentes et al. (2022)), it is shown the implementation of a device for AQ outdoor evaluation using directly this module without calibration, to map AQ pollutants in a metropolitan area. In (Felici-Castell et al. (2023)), this module is used in an AQ monitoring network, where different neural networks have been trained for forecasting of pollutant concentrations, with an estimation error of 7.2% on average and where the calibration process is done on a daily basis, but not specified. In (Vaheed et al. (2022)), this module is used for indoor AQ monitoring and calculating an AQ index. In (Antonenko et al. (2023)), the authors explain briefly the use of a neural network to determine (classify) types of air: with or without pollution. Also, in (Kennedy et al. (2021)), it is shown a prototype to measure ground to stratosphere AQ using this module in a drone. However, the variability among the individual sensors is high, stressing that the calibration process is complex and it has not been done.

Regarding LCS performance analysis, the authors in (Borrego et al. (2016)) conducted a two-week assessment in Aveiro (Portugal) of various LCS models. Specifically for $O_3$, the best performance compared to a reference station was achieved by the MiCS-OZ-47 and Alphasense B4 Electrochemical sensors, which obtained a coefficient of determination $R^2$ values (and MAE in ppb) of 0.77 (7.66) and 0.70 (2.4), respectively.

The calibration process of these LCS is a challenge as mentioned before, where ML and Deep Learning (DL) models can be used. In (Wang et al. (2024)), a low-cost multi parameter AQ system based on $PM_{2.5}$, $PM_{10}$, $SO_2$, $NO_2$, $CO$ and $O_3$, along with Temp and RH is proposed using and evaluating various calibration algorithms. For $O_3$, the algorithms are ranked from best to worst fit as follows: RF, K-Nearest Neighbors (KNN), Back Propagation (BP), Genetic Algorithm Back Propagation (GA-BP), and Multiple Linear Regression (MLR), with $R^2$ values (MAE, in $\mu g/m^3$) of 0.98 (2.88), 0.87 (7.33), 0.83 (11.14), 0.83 (10.90), and 0.74 (13.46), respectively. With a mean $O_3$ concentration of approximately 70 $\mu g/m^3$, as shown in their Figure. 12, the RF model achieves a MRE of 4.11%. In (Cavaliere et al. (2023)), based on $O_3$ and $NO_2$ metal oxide sensors, along with Temp and RH, the authors analyzed different calibration options using uni-variate/multi-variate, linear/non-linear and parametric/non-parametric approaches with algorithms such as Linear Regression (LR), Non-Linear Regression (NLR), Support Vector Machines (SVM), RF and GB. They concluded that Multiple Random Forest (MRF) achieved the highest accuracy during Phase I (pre-deployment), with an $R^2$ of 0.98 and MAE (MRE) of 4.31 (5.74%), considering a mean $O_3$ during their deployments of 75 $\mu g/m^3$, depicted in their Figure 3 and 7. However, in Phase II (field validation) conducted at a different location, the performance worsened, with the MAE (MRE) 22.22 (29.62%) while MLR 12.96 (17.28%). In this case, MLR provided better results. The authors conclude that MLR may be a more suitable solution for representing physical models beyond the Phase I calibration dataset, demonstrating better transferability across diverse spatial and temporal settings, highlighting that parametric models such as MLR have a defined equation with only a few parameters, making them easier to adjust for potential changes over time. In (Wang et al. (2020)), the authors propose a category-based calibration approach (piecewise) using ML, which builds separate regression models for different pollutant concentration levels. This proposal is tested on $CO$ and $O_3$ data from two Chinese cities, Fuzhou and Lanzhou, with good and bad AQ, with mean $O_3$ concentrations of 69.545 $\mu g/m3$ and 49.781 $\mu g/m3$ respectively for 11 months (48 weeks). The achieved metrics for the best results are given by Extreme GB and Light GB machine algorithms (outperforming linear regression and RF) with MAE ($\mu g/m3$) of 10.75 and 10.98 in Lanzhou city respectively, and 13.83 and 14.98 in Fuzhou city, with a MRE greater than 19.88%. In (Zimmerman et al. (2018)), the authors show calibration models (using 16 weeks data) to improve sensor performance, highlighting that RF approach is more robust since it accounts for pollutant cross-sensitivities. Using specific LCS (RAMP system), they achieve an MRE of 15% for $O_3$. In the study performed by (Johnson et al. (2018)), the calibration of an aerosol sensor for $PM_{2.5}$ is carried out by comparing simple linear regression models with GB using the PPD42 PM sensor (Shinyei (2024)). The study concludes that gradient boosting performed better and significantly improved the performance of the sensors, reaching a $R^2$ of up to 0.76. In (Casey et al. (2019)), the authors show that Neural Networks (NN) generally outperform lineal models to quantify $O_3$, $CO$, $CO_2$, and $CH_4$ in ambient air, using gas sensors integrated into U-Pod AQ monitors. Besides, they highlight that NN capture the complex nonlinear interactions among multiple gas sensors, considering factors such as Temp, RH and atmospheric chemistry. Also, in (Esposito et al. (2016)), the authors use dynamic NN for calibration achieving models with $R^2$ (MAE) (in ppb) of 0.69 (7.45), with a MRE of 42%.

In this context, when using AI techniques on environmental research, it is important to follow the recommendations and good practices given by (Zhu et al. (2023)) based on a review of more than 148 highly cited research papers. In this paper, it is highlighted that data preprocessing, analysis and interpretability are often overlooked, such as Feature Importance Analysis

(FIA), Principal Component Analysis (PCA) and Feature Selection (FS) as part of the Exploratory Data Analysis (EDA). A good example of the use of these good practices is shown in (Cavaliere et al. (2023)). In addition, in (Zhu et al. (2023)), it is said that the process of optimizing algorithms through the selection of their hyperparameters (Hyperparameter Optimization (HPO)) is neglected in most of the environmental research studies considered. For instance, in (Johnson et al. (2018)), better results are obtained with GB, but HPO is not performed in the model, which could allow further improvements of the results. In (Malings et al. (2019), (Wang et al. (2020)) and (Zimmerman et al. (2018)), it is taken into account some aspects related to the data analysis focused on the optimization of the problem, but they do not carry out a HPO. In (Esposito et al. (2016)), the authors carry out a kind of simple HPO, based on raw tests of different architectures and modifying hyperparameters, such as the number of hidden layers of the model, tapped delay length and feedback delay line length, concluding that a dynamic approach to these parameters improves the results with respect to a static approach without changing the value of these parameters.

Regarding the selection of parameters, in (Johnson et al. (2018)), the authors does not perform an analysis using techniques such as the aforementioned FIA and FS, but a sensitivity analysis using different meteorological variables (such as Temp and RH), determining that it is useful information for GB. In (Malings et al. (2019)), the quantification of the importance of the model variables is mentioned as a mean to understand which information is useful, concluding that for RF, to add additional information apart from AQ measurements, such as Temp and RH are very helpful. In (Esposito et al. (2016)) and (Wang et al. (2024)), the authors do not include a specific analysis of the relative importance of different variables or features. However, a good example of FS is depicted in (Okafor et al. (2020)), where it is shown that identifying the environmental factors affecting LCS is crucial for improving data quality using data fusion and ML. These factors are then incorporated into the development of the calibration model.

In conclusion, in order to increase the resolution of city-scale AQ monitoring according to the recommendations given by (Directive 2008/50/EC (2008)) as mentioned before, it is necessary to perform a calibration process of these LCS. In this scenario, we focus on $O_3$ calibration using ensemble ML techniques to minimize inaccuracies and nonlinearities, comparing four different models, considering different environmental variables as well as metadata mainly to account for sensor aging effects. For this purpose, it is necessary to carry out a thorough data treatment with a good practice criteria (Zhu et al. (2023)) including HPO, FIA and/or FS, which are usually overlooked. In a scenario with low $O_3$ concentration, we achieve interesting results compared with the related work, as shown in Section 4.

## 3 Building the dataset and using Machine Learning algorithms

In this section we explain the process to gather AQ monitoring information from a prototyped low-cost Internet of Things (IoT) node based on the ZPHS01B AQ module, how it is deployed and how the datasets are built to apply ML techniques for calibration purpose. For this, we generate two datasets in the city of Valencia (Spain), at two different locations with the same characteristics (close to the ring road but separated by 4.1 km), covering periods of 165 and 239 days.

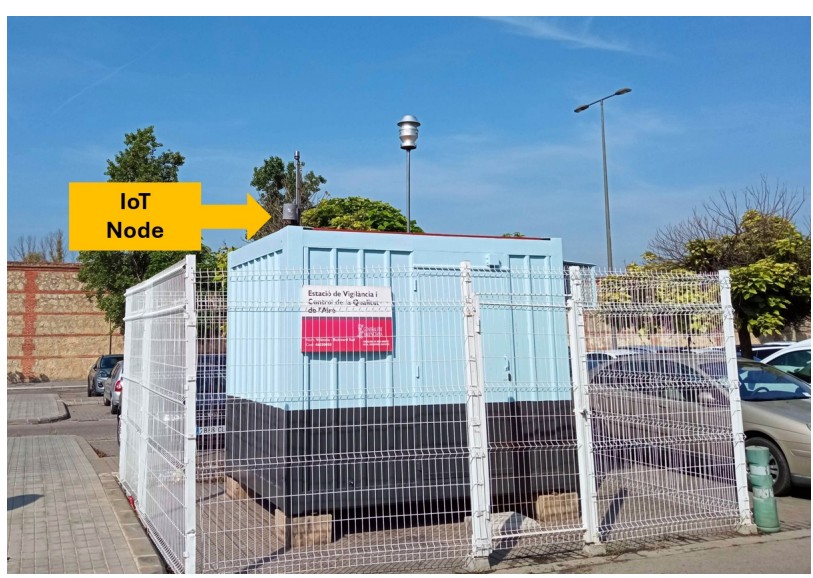

**Figure 1.** Detail of the standardized AQ monitoring station and the AQ node with a ZPHS01B module located at Bulevar Sur (Valencia, Spain).

## 3.1 Building the datasets

To calibrate the $O_3$ sensor from the ZPHS01B module, we require a dataset (named dataset-1) to train various ML models. For this purpose, we use as reference values, $O_3$ concentration readings from the standardized AQ station in the Valencian AQ Monitoring Network (VAQMN), at Bulevar Sur (Valencia, Spain) managed by Generalitat Valenciana (GVA) with latitude and longitude 39.450389 and -0.396324, respectively, as shown in Figure 1. In this picture, the IoT node is pointed out, placed 4 meters above ground level in accordance with Directive 2008/50/EC (Directive 2008/50/EC (2008)). These reference values are given in $\mu g/m^3$ periodically averaging every 10-minutes. The AQ station data is retrieved from (Generalitat Valenciana (2025a)). The dataset-1 includes 165 days, from June $8^{th}$ 2023 till November $20^{th}$ 2023. The ZPHS01B module's readings are taken at a rate of 10 samples per minute, one sample every 6 seconds. Notice that, as a first approach, creating a dataset with different locations is not recommended, as it could alter environmental conditions and interfere with the training process. That it is the reason we generate to different and independent datasets.

Table 3 presents the structure and main statistics of the dataset-1. The units used for $O_3$ concentration from the standardized and regulated station are in $\mu g/m^3$, meanwhile in the ZPHS01B module are in ppm. Both units are typically used in a formal and academic context, but we need to standardize them. The formula used to carry out this conversion for $O_3$ is in standard conditions: "Concentration ($\mu g/m^3$) = molecular weight (48 g/mol) x concentration (Parts Per Billion (ppb)) ÷ 24.45", that is 1 ppb is 1.96 $\mu g/m^3$ (Breeze Technologies (2024)).

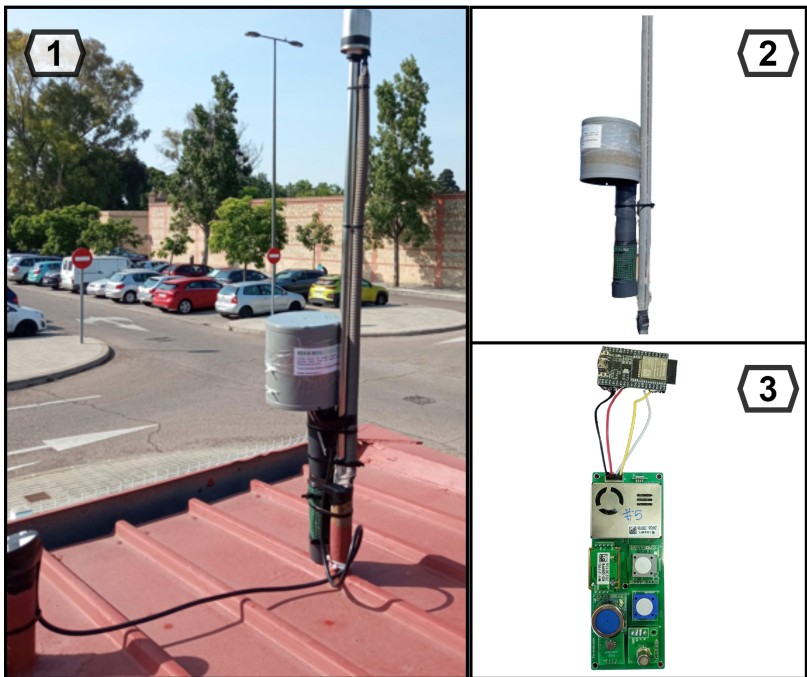

**Figure 2.** (1) AQ IoT node; (2) deployment detail; (3) hardware detail

In Figure 2, it is shown the IoT node (and its housing) that keeps the ZPHS01B module within a PVC pipe with a small fan at the top, to ensure air circulation. In the head of this node, it is placed the microcontroller that sends data via the LTE-M communications. Further detail about the features of this node is given in (Meneses-Albala et al. (2025)).

In addition, in order to test the proposed models in this paper and their generalization in Section 4, we have used another dataset (named dataset-2) with two different AQ IoT nodes (Node 1 and 2), from the standardized AQ monitoring station called *Moli del Sol* (Valencia, Spain) with latitude and longitude 39.48113875, -0.40855865. This station is 4.1 km away from the previous one. Its data is retrieved from (Generalitat Valenciana (2025b)). In this case, this dataset is from May 31, 2024 till January 25, 2025, with 239 days. Now on, we will refer always to dataset-1 as the dataset, except in Section 4 where we generalize the models with dataset-2.

### 3.2 Analyzing the dataset

The initial data collection (dataset-1) is based on 6-second frequency samples. Based on this collection, three datasets have been created by averaging data over different time monitoring intervals: 10 min., 30 min. and 1 h. with 23496, 7843 and 3922 samples respectively. The lowest 10 min. interval is given by the standardized AQ station and 30 min and 1 h are common time base for AQ parameters. Although they are not large data-set, it is sufficient as shown in (Zhu et al. (2023)), due to the relationship (ratio) between sample size and feature size, 4 features in total as seen next. This ratio is called, Sample-size to Feature-size Ratio (SFR), being recommended a SFR higher than 500. More detail is given in Section 3.3.

Initially, the datasets were cleaned of invalid data. Notice that from the readings of the standardized AQ station, we had 275 Not a Number (NaN) during this period, that in our case were replaced using the quadratic interpolation method, since experimentally it gave better results and made the interpolation closer to the ozone signal. This explanation to prepare the dataset, also known as Missing Data Management (MDM), is recommended according to (Zhu et al. (2023)).

**Table 3.** Summary of main statistics of the Dataset-1: Minimum (Min.), Maximum (Max.), Mean (Mean), Standard Deviation, Median Absolute Deviation (MAD), percentage of samples taking Different values (Diff.) and High correlation (High corr.)

|  | Temp [ºC] | RH [%] | PM$_{2.5}$ [$\mu g/m^3$] | CO$_2$ [ppm] | NO$_2$ [$mg/m^3$] | CO [$mg/m^3$] | CH$_2$O [$mg/m^3$] | TVOC [levels] | O$_3$ [$\mu g/m^3$] | O$_3$ref [$\mu g/m^3$] |
|---|---|---|---|---|---|---|---|---|---|---|
| **Min** | 5.24 | 62.29 | 21.25 | 693.43 | 0.78 | 0 | 0.005 | 0 | 39.57 | 8.71 |
| **Max** | 42.26 | 118 | 83.69 | 1792.50 | 18.81 | 0.75 | 1.21 | 2.95 | 255.76 | 97.85 |
| **Mean** | 20.60 | 91.31 | 49.99 | 780.33 | 15.27 | 0.34 | 0.021 | 0.024 | 114.39 | 55.72 |
| **SD** | 5.70 | 18.12 | 18.14 | 57.16 | 5.65 | 0.28 | 0.02 | 0.13 | 67.11 | 24.83 |
| **MAD** | 3.92 | 16.37 | 13.31 | 24.53 | 0.59 | 0 | 0.001 | 0 | 51.40 | 16.21 |
| **Diff.** | 99.1% | 81.9% | 87.9% | 97.5% | 50.6% | 0.2% | 81.2% | 5.8% | 75.0% | 30.3% |
| **High corr.** | yes | yes | yes | yes | not | yes | not | not | yes | yes |

Table 3 shows a summary of main statistics of the dataset-1. For each parameter is shown: the Minimum value (Min.), Maximum value (Max.), Mean value of all entries (Mean), Standard Deviation, Median Absolute Deviation (MAD), percentage of samples taking Different values (Diff.) and High correlation (High corr.) with others.

From these results, it is worth mentioning that the $CH_2O$, $CO$, $TVOC$ and $NO_2$ sensors do not seem to be working properly in the ZPHS01B module. $CH_2O$, $CO$ and $TVOC$ are almost always stuck to values close to zero, seeming not to excite at normal concentrations, with very low variability. On the other hand, the $NO_2$ sensor appears saturated. Thus in practice, the number of used features from Table 3 are 5, that is from the initial 9 (the reference is not included), we remove these 4 ($CH_2O$, $CO$, $TVOC$, and $NO_2$). Also, RH sensor has a positive offset as we can see from the maximum value, 118%.

Figure 3 shows the $O_3$ readings in $\mu g/m^3$ from the LCS and the regulated station (reference) for one week. It can be seen that there is an offset in the LCS readings over the ones from the reference. Also, it is clear how the $O_3$ LCS captures the trends, useful information for the ML models. A further analysis of these sensor readings in the frequency domain is shown in Appendix A, where a repeated daily pattern is clearly observed, as expected based on how $O_3$ is generated from other pollutants produced by road traffic and complex photochemical reactions, as discussed in Section 1.

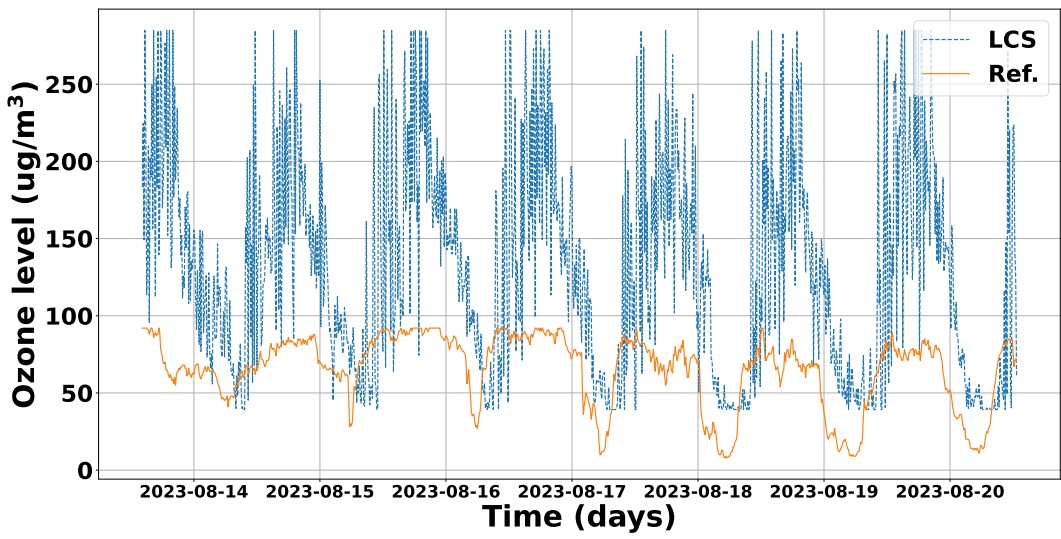

**Figure 3.** $O_3$ readings in $\mu g/m^3$ from the LCS and Reference for one week

### 3.3 Feature Importance Analysis and Selection

FIA and FS play crucial roles in ML models, especially in environmental research, by helping to preserve essential features (variables), reduce noise and enhance model efficiency, particularly relevant when dealing with a small set of samples or large numbers of variables (Zhu et al. (2023)).

**Table 4.** FIA of ozone's complementary parameters for Random Forest (RF), Gradient Boosting (GB), Adaptive Boost (ADA) and Decision Tree (DT), in bold the selected ones, contribution higher than 0.8.

| Model | Temp | RH | PM$_{2.5}$ | CO$_2$ | NO$_2$ | O$_3$ref | CO | TVOC | CH$_2$O |
|---|---|---|---|---|---|---|---|---|---|
| RF | **0.128** | **0.103** | 0.069 | **0.222** | 0.078 | **0.269** | 0.002 | 0.003 | 0.064 |
| GB | **0.107** | **0.105** | 0.052 | **0.211** | 0.057 | **0.253** | 0.001 | 0.001 | 0.068 |
| ADA | **0.119** | **0.097** | 0.064 | **0.246** | 0.067 | **0.287** | 0.001 | 0.001 | 0.066 |
| DT | **0.115** | **0.088** | 0.070 | **0.232** | 0.061 | **0.276** | 0.001 | 0.002 | 0.061 |

Table 4 shows the normalized output of the FIA using the *scikit-learn* library (Pedregosa et al. (2011)), for the parameters complementary to $O_3$, for each ML models used. In order to determine the most useful parameters for the models, a threshold is established in 0.08, that is 8% of importance. These parameters are in bold. Notice that the set of parameters with the highest importance, is repeated for all models: Temp, RH, $CO_2$ and $O_3$.

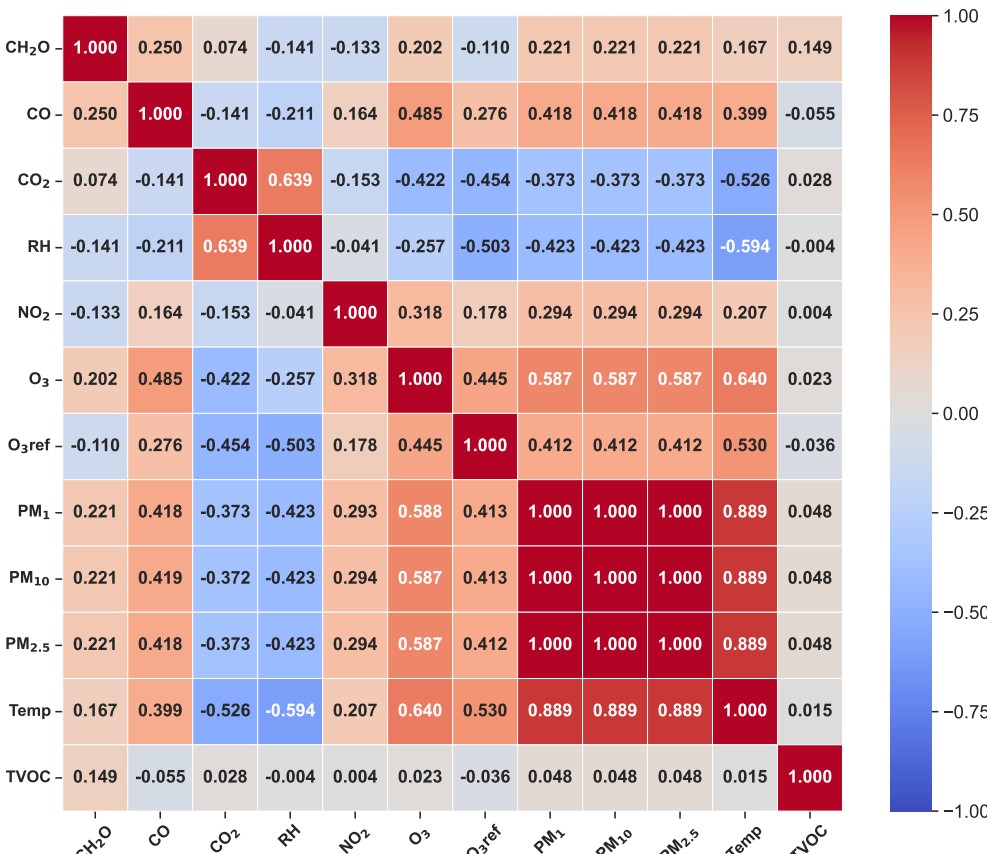

**Figure 4.** LCS readings and $O_3$ reference correlation matrix

225 With regard to FS, Figure 4 shows the correlation matrix among these variables. There is a high correlation among all $PM_x$ readings because all of them are calculated directly from the $PM_{2.5}$ (Zhengzhou Winsen Electronics Technology Co. (2024)). Also, from this analysis, we stress that Temp and RH, are the best correlated with the rest of variables, as well as $O_3$ LCS, $O_3$ reference, $PM_{2.5}$ and $CO_2$, but these ones with a lower correlation. Thus, all this information is very valuable to train the ML models.

230 **3.4 Applying Machine Learning algorithms**

As mentioned before in environmental research, the use of ML algorithms, in particular ensemble models, has increased significantly compared to DL (Zimmerman et al. (2018)). Some of the most popular ensemble algorithms are RF or GB related models (Obregon and Jung (2022)). Furthermore, based on our experience, we recognize that in AQ monitoring scenarios using LCS such as the ZPHS01B module, datasets are often limited and constrained, which affects the use of DL techniques, as they

235 usually tend to overfit.

This paper evaluates these ensemble ML algorithms: RF, GB and ADA algorithms, implemented in the *scikit-learn* (Pedregosa et al. (2011)) library (in the ensemble submodule), that offers efficient solutions for time series regression problems as this one. These evaluated methods exhibit the ability to handle non-linear relationships and adapt to changing patterns over time. In addition, the DT model, belonging to the tree submodule of *scikit-learn*, is also evaluated, since it is a common base of this type of ensemble algorithms.

To optimize these models as indicated in (Zhu et al. (2023)), there are different techniques and tools in order to carry out the HPO, being *GridSearch* (Pedregosa et al. (2011)) the most commonly used method to obtain a good configuration for these algorithms. *GridSearch* in *scikit-learn* is a hyperparameter tuning technique that exhaustively searches through a user-defined hyperparameter space to find the optimal combination for a ML model. These hyperparameters are external specific model configurations settings. It systematically evaluates the model's performance across all possible user-defined hyperparameters using cross-validation, aiming to identify the configuration that maximizes estimation accuracy or minimizes a specified loss function. We choose this method due to its higher flexibility compared to other tools such as *RandomSearch* (Pedregosa et al. (2011)) that has a more random approach.

Next, we discuss the different supervised ML algorithms used and the selection of the different hyperparameters taking into account the best results of $R^2$, Root Mean Square Error (RMSE) and MAE.

### 3.4.1 Random Forest (RF)

**Table 5.** RF hyperparameters evaluated on *GridSearch* showing in **bold** the combination that gives the best results in terms of $R^2$, RMSE and MAE.

| No. of estimators | Max. depth | Max. features |
|---|---|---|
| 50, **100**, 250, 500, 900 | 2, 5, 7, **None** | sqrt, log2, 0.1, 0.3, 0.5, **1.0** |

RF is an ensemble algorithm that relies on constructing multiple DT during training. Each tree is trained on a random subset of the dataset, and the final predictions are obtained by averaging the individual predictions for all of them. This "forest" approach helps to mitigate overfitting and improves the model's generalization. Furthermore, introducing randomness in the selection of features and samples during tree construction contributes to a more robust and accurate model for regression tasks. Table 5 shows the hyperparameters evaluated, in bold the best option. The *number of estimators* refers to the number of trees in the forest, while the maximum depth refers to the *maximum depth* of the tree. The *maximum features* variable determines the upper limit on the number of features to consider when splitting a tree into two child nodes during the tree construction process. Note that as the *number of estimators* does not have a significant role in this use case, we use the default value, 100.

**Table 6.** GB hyperparameters evaluated on *GridSearch* showing in **bold** the combination that gives the best results in terms of $R^2$, RMSE and MAE.

| No. of estimators | Max. depth | Max. features |
|---|---|---|
| 50, 100, 250, 500, **900** | 2, 5, 7, **None** | sqrt, log2, 0.1, 0.3, 0.5, **1.0** |
| **Learning rate** | **Subsample** | **Loss** |
| 0.01, 0.05, **0.1**, 0.3 | 0.5, 0.8, **1.0** | **squared err.**, absolute err., huber |

### 3.4.2 Gradient Boosting (GB)

GB is an ensemble algorithm based on the iterative construction of weak DTs, which are sequentially aggregated to enhance the predictive capability of the model. In each iteration, it focuses on correcting the residual errors of the existing model by fitting a new DT to capture the deficiencies of the current model. The weighting of individual trees is determined by a learning rate, and the final output of the model is the weighted sum of predictions from all these trees. This gradual building process and the ability to handle nonlinear relationships in the data make GB effective for regression tasks. Table 6 shows the hyperparameters evaluated, in bold the best option. In addition to the previous hyperparameters, in this case, the *loss* hyperparameter refers to the loss function to be optimized, while *learning rate* reduces the contribution of each tree according to the value of the variable. The *subsample* hyperparameter represents the fraction of samples that will be used to adjust the individual base learners and if it is less than 1.0, it results in Stochastic Gradient Boosting (SGB).

### 3.4.3 Adaptive Boosting (ADA)

**Table 7.** ADA hyperparameters evaluated on *GridSearch* showing in **bold** the combination that gives the best results in terms of $R^2$, RMSE and MAE.

| No. of estimators | Learning rate | Loss |
|---|---|---|
| **50**, 100, 250, 500, 900 | **0.01**, 0.05, 0.1, 0.3 | linear, square, **exponential** |

ADA is an ensemble algorithm, that its primary goal is to improve the predictive accuracy by combining multiple weak regression models. When training, ADA assigns weights to data instances, giving more emphasis to observations that were poorly predicted in previous iterations. Its construction involves the sequential aggregation of regression models, each fitted to correct errors from the existing combined model. The final model is a weighted combination of individual predictions from the base models. ADA is particularly effective in enhancing generalization capability and reducing overfitting in regression tasks. Table 7 shows the hyperparameters evaluated, in bold the best option. In this model, there is a key concept to run the optimization process related to the *estimator* variable, that by default is an instance of type *DecisionTreeRegressor*, initialized

with a maximum depth value of three. If the value of this hyperparameter is not modified, this model is largely constrained. Also, notice that as the number of estimators does not have a significant role on this use case, we use the default value of 50 estimators. The other hyperparameters have the same meaning in this model.

### 3.4.4 Decision Tree (DT)

**Table 8.** DT hyperparameters evaluated on *GridSearch* showing in **bold** the combination that gives the best results in terms of $R^2$, RMSE and MAE.

| Max. depth | Max. features | Splitter |
|:---:|:---:|:---:|
| 2, 5, 7, **None** | sqrt, log2, 0.1, 0.3, 0.5, **1.0** | **best**, random |

DT is an algorithm that recursively partitions the dataset based on features, aiming to create a hierarchical structure of decision nodes to make predictions. Table 8 shows the hyperparameters evaluated, in bold the best option. The *splitter* hyperparameter indicates which strategy is used to perform the splitting at each node.

## 4 Results

We evaluated the performance metrics of these ML models under different configurations (in terms of $R^2$, RMSE, MAE in $\mu g/m^3$ and Mean Absolute Percentage Error (MAPE) and execution time in seconds), with the optimized hyperparameters that achieve higher $R^2$ and lower errors. Also, we used the three different datasets given by different monitoring intervals: 10 and 30 min and 1 h, as depicted in Section 3.2. We tested different training-test ratio percentages from these datasets: 60%-40%, 70%-30%, 80%-20% and 90%-10%, denoted as 60/40, 70/30, 80/20 and 90/10. Note that when we split the dataset for training and testing, both sets remain independent and isolated. However, during the training process itself, the dataset is further divided into two parts: one for training and the other for validation. By default, we allocate 80% of the data for training and 20% for validation. In this process, the training and validation datasets are combined across different iterations. From all of them, we have achieved the best results in terms of these performance metrics with 90/10 training-test ratio with a monitoring interval of 10 min, as shown in Table 9. Besides, from the analysis carried out in Section 3.3, for the feature selection, we proceed in this section with the features that provide also the best results, based on [date, $O_3$, Temp, RH]. Notice that "date" is included as metadata to account for aging effect and improve the models following traffic pattern. We see that fewer features, better results, i.e. increasing the SFR. Then, other dimensionality reduction techniques are not required. If we add more features that are not so significant, it makes the dataset poorer. Notice that the performance metrics shown in Table 9 are the weighted average of each metric over 100 different iterations by changing the content of the training and test set to obtain results with the minimum bias as possible.

During the training process, we can observe the convergence of the performance metrics that provides information about overfitting, considering both the training and validation datasets. In Appendix B, it is included this information analyzing both

$R^2$ and RMSE across different iterations. Each model uses its own reference hyperparameter for convergence. In particular, in Figure B1, we observe the fit of the model in terms of $R^2$, with a better fit with training than with validation, as expected, similarly as we can see for RMSE in Figure B2. It should be noted that the convergence process with training does not reach a perfect fit in any case, which justifies and supports initially the conclusion that there is no overfitting in the models. Moreover, we see that the achieved $R^2$ and RMSE scores for both training and validation are better than the values shown for testing in Table 9, because the testing dataset does not participate in the training process.

**Table 9.** Performance metrics for HPO models with 90/10 and 80/20 (training/testing) ratio

| Model | GB | | RF | | ADA | | DT | |
|---|---|---|---|---|---|---|---|---|
| Ratio | 90/10 | 80/20 | 90/10 | 80/20 | 90/10 | 80/20 | 90/10 | 80/20 |
| $R^2$ | 0.938 | 0.936 | 0.927 | 0.924 | 0.922 | 0.920 | 0.878 | 0.863 |
| RMSE | 6.492 | 6.664 | 7.093 | 7.253 | 7.289 | 7.416 | 9.149 | 9.735 |
| MAE | 4.022 | 4.221 | 4.185 | 4.415 | 3.642 | 3.833 | 4.684 | 5.104 |
| MAPE | 0.194 | 0.206 | 0.208 | 0.228 | 0.160 | 0.175 | 0.206 | 0.226 |
| Time | 66.937 | 61.054 | 18.316 | 16.618 | 7.805 | 7.078 | 0.212 | 0.194 |

It is worth mentioning that the improvement achieved by HPO is greater in GB and ADA models than in RF and DT, which are already well-optimized with default values. In particular, for the optimized GB and ADA models, $R^2$ is improved by 42% and 182%, respectively, while RMSE is reduced by 57% and 66%. However, the execution time required for training is influenced by HPO, increasing to 66.937s and 7.805s for GB and ADA, respectively, as shown in Table 9. We highlight that RF and DT are already well-optimized, and their execution times remain unchanged between the default and optimized versions.

In Figure 5 it is shown the calibration process for both the default and HPO models vs $O_3$ reference given by the different algorithms.

However, it is common to use a 80/20 training/test ratio (Zhu et al. (2023)). For this ratio, Table 9 also shows these results with the optimized models by HPO, where the GB model is the best one again, as it happened with previous 90/10 ratio.

A summary of these metrics ($R^2$ and errors) for the GB model, with different monitoring intervals and different training/test ratio percentages are shown in Figure 6. We can see how increasing the training %, the trend is to improve the accuracy of the model ($R^2$ getting closer to 1) and to reduce lightly the errors but increasing the training time, as it could be expected. Similar behaviors are exhibited by the other models, particularly by the ADA model. Regarding overfitting, Figure 6 shows that the error difference between using 90% and 60% of the data for training (the maximum and minimum percentages, respectively) is approximately 2% in the worst-case scenario (1-hour dataset). This suggests that overfitting is not significant in the proposed model, as we mentioned before during the convergence process shown in Appendix B.

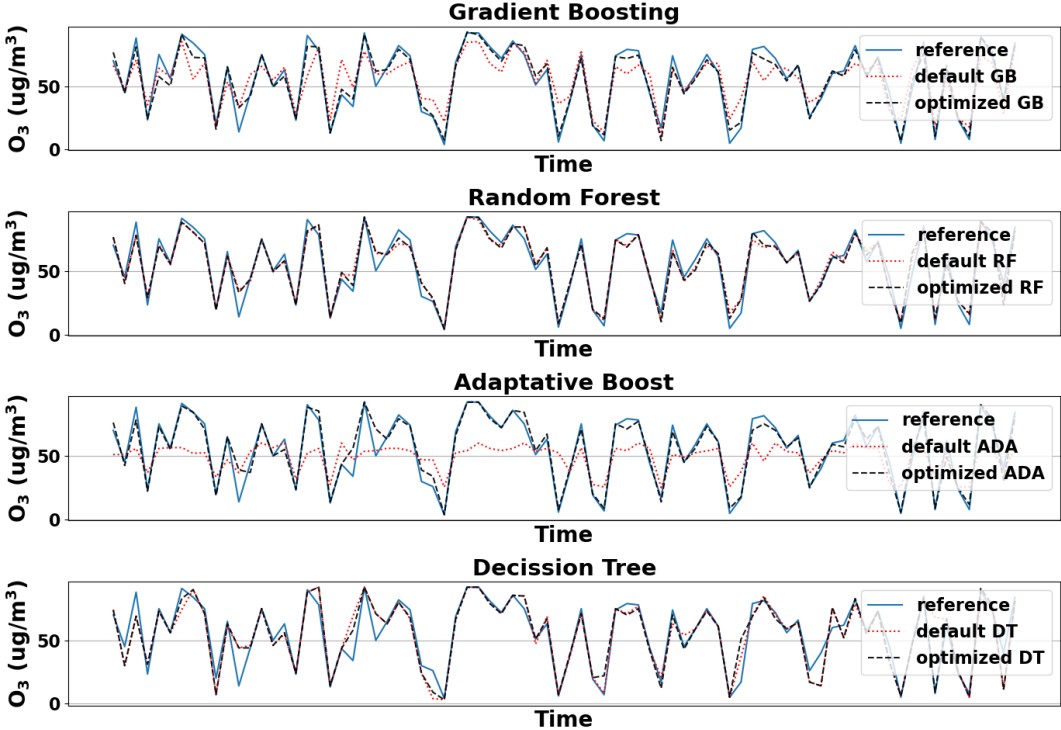

**Figure 5.** Ozone calibration done with default and optimized models with 90/10 (training/testing) ratio dataset

In terms of generalization as mentioned in Section 3.1, we have checked the same proposed models with dataset-2 under the same conditions, with 90/10 (training/testing) ratio. In Figure 7, we summarize the performance metrics given by the best model based on GB for dataset-1 and for Node 1 and 2 from dataset-2 respectively. In particular, if we focus on MAE, we see that Node 2 performs slightly better than Node 1 in dataset-2, likely due to manufacturing variations associated with their 330 low-cost, as well as the results from dataset-1 are between these two, validating its generalized behavior. In terms of RMSE, the results from dataset-2 with both nodes is slightly better since it is larger. In all these cases, $R^2$ is higher that 0.938.

**Table 10.** Improvement (in%) of $O_3$ calibration from the raw readings with the different optimized models.

|  | GB | RF | ADA | DT |
|---|---|---|---|---|
| $\mathbf{R^2}$ | 258.13% | 256.27% | 256.1% | 246.27% |
| **RMSE** | 93.05% | 92.43% | 92.29% | 89.85% |
| **MAE** | 94.05% | 93.82% | 94.59% | 92.79% |
| **MAPE** | 62.75% | 58.8% | 68.35% | 59.12% |

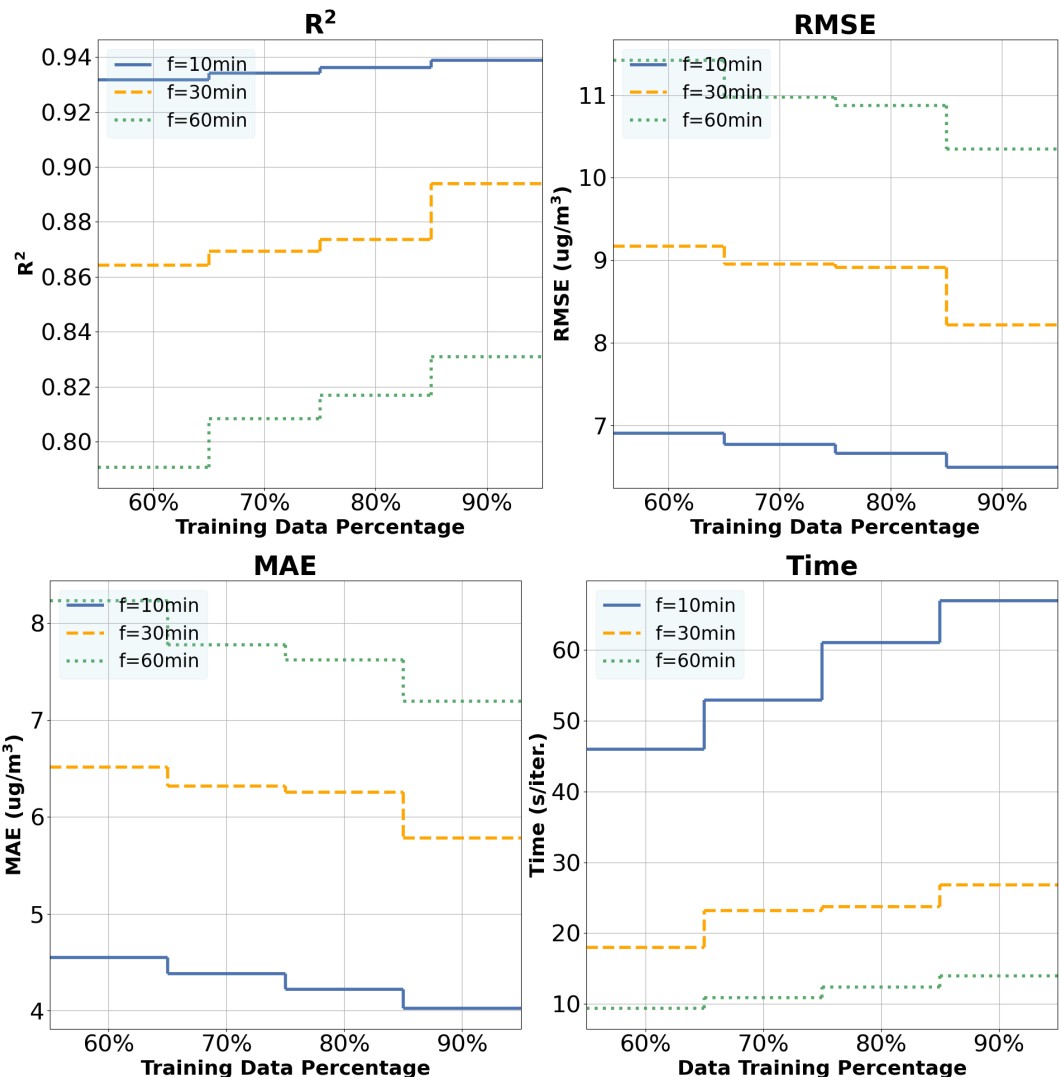

**Figure 6.** Ozone estimation analysis for GB default and optimized models with different % training datasets and monitoring intervals

In Table 10, we show the improvement in % using the different ML models for the calibration process from the LCS raw readings of the module, highlighting the better performance of GB model compared to the other models. Notice that with this model, GB, the initial MAE from the raw readings was 67.59 $\mu g/m^3$ reducing it to 4.022 $\mu g/m^3$, that is an improvement of 335  94.05% as depicted in this table.

Finally, in Table 11, we compare our models for $O_3$ calibration for LCS, against the related work with a similar approach, highlighting the location, platform (and sensors used), $\mathbf{R^2}$, MRE along with additional comments about the detail of the models used and dataset duration. First, we must stress that the starting point of the selected papers is slightly different compared to ours, since these studies have used more reliable and expensive LCS, approximately ten times more expensive that the

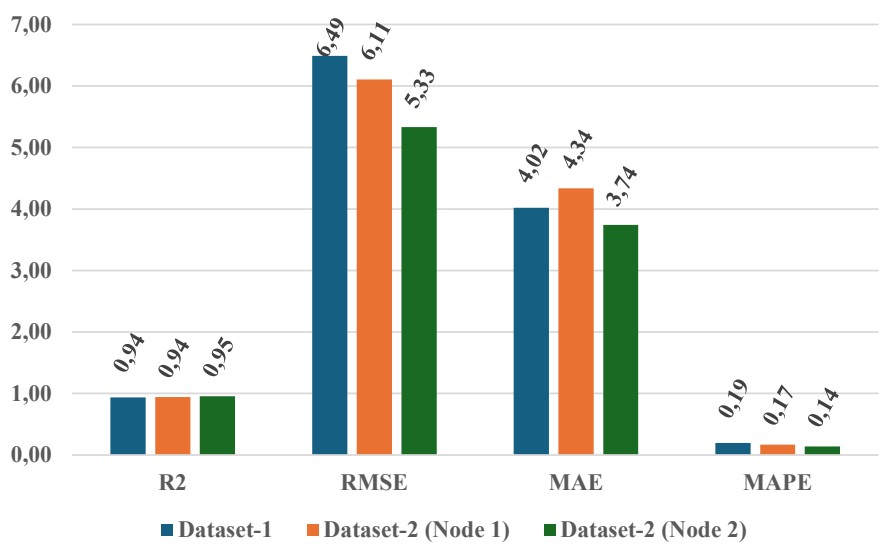

**Figure 7.** Performance metrics comparison using GB$_{optimized}$ algorithm with 90/10 (training/testing) ratio over dataset-1 and dataset-2 (Node 1 and 2).

ZPHS01B module. Moreover, since ML-based algorithms show the best results as discussed in Section 2, we have focused exclusively on them, evaluating up to four different models, whereas other studies have only considered one or two. Our model, in particular GB with 4 features (including "date" as metadata), as shown in Section 3.3, achieves a MRE of 7.21% (given by MAE 4.022 $\mu g/m^3$ with 90/10 dataset (Table 9) and the mean $O_3$ value of 55.72 $\mu g/m^3$ (Table 3)). Besides, not all of these works follow and discuss an structured EDA with FIA, FS and HPO. In particular, when compared to the first two

works with slightly better results, in (Wang et al. (2024), we appreciate higher $O_3$ values, mean values higher than 70 $\mu g/m^3$, while in our case we have lower levels (55.72 $\mu g/m^3$)), as well as there is not a complete EDA. It is important to note that these sensors perform worse at low concentrations than at high ones due to their sensitivity limitations and the weakness of the signals generated, as well as interference from other pollutants. Finally, in (Cavaliere et al. (2023)), although the authors use a complete EDA, they only use two sensors ($NO_2$ and $O_3$) apart from Temp and RH, and the aging effect is considered a

posteriori, while this information is included in our case by date in our models, which also detects other patterns derived from road traffic.

## 5 Conclusions

This paper focuses on ground-level ozone ($O_3$), as it serves as an indicator of other pollution levels in urban areas using LCS nodes based on the ZPHS01B module. These nodes will enable an increase in the spatial sampling of AQ monitoring in cities,

| Study | Location | Platform, Sensor | $R^2$ | MRE [%] | Comment |
|-------|----------|------------------|-------|---------|---------|
| (Wang et al. (2024)) | Zhengzhou (China) | by Hanwei Electronics Corp, $O_3$ B4 Alphasense | 0.93 | 4.11 | 52 weeks dataset with RF and HPO |
| (Cavaliere et al. (2023)) | Florence, Montale (Italy) | AirQino LC, $NO_2$ MiCS-2714, $O_3$ MiCS-2614 | 0.98 with MRF | I:MRF 5.74. II:MLR 17.28, MRF 29.62 | 61 weeks dataset with MRF and MLR, using complete EDA |
| (Wang et al. (2020)) | Lanzhou (China) | Sailhero instrument, - | - | 19.88 | 48 weeks dataset, category-based calibration (piecewise) with Extreme GB and FS |
| (Zimmerman et al. (2018)) | Pittsburg (USA) | RAMP, Alphasense Ox-B431 | 0.86 | 15 | 16 weeks dataset with RF |
| (Esposito et al. (2016)) | Cambridge (UK) | SnaQ, Alphasense B4 Electrochemical | 0.69 | 42 | 5 weeks dataset using Dynamic NN with a kind of HPO |
| **Our model** | Valencia (Spain) | ZPHS01B, Winsen ZE27 | 0.93 | 7.21 | 57 weeks dataset using GB with FIA, FS and HPO |

**Table 11.** Comparison with similar related works

following the interest of AQG (Organization et al. (2021)) and in line with the future plans of the related directives, ideally at least one sample per 100 $m^2$, according to Annex III-B of the European (Directive 2008/50/EC (2008))..

Given the low accuracy and nonlinearities of these nodes' sensors, we employed ML models (particularly DT and the ensemble algorithms (GB, RF and ADA)), after thorough data analysis, considering additional environmental information and including metadata to account for the aging effect and detect other patterns derived from road traffic, we reduce the estimation error by approximately 94.05%, and more than 89% in the other models. In particular, using the GB algorithm, we achieve a MAE of 4.022 $\mu g/m^3$ and a MRE of 7.21%, outperforming related work while using a module approximately 10 times less expensive.

Initially, we used a dataset spanning 165 days (with low $O_3$ concentrations, and a mean value of 55.72 $\mu g/m^3$), with different monitoring intervals, giving the best results when we use 10 min monitoring interval, as it could be expected. If we use higher monitoring intervals (30 min or 1 hour), we see that we start losing details, smoothing the dataset and overlooking different behaviors that in the ML process helps to reduce the prediction error. For the training process, we have carried out several techniques (FIA and FS) in order to select the most relevant features, applying HPO within the different models, with different percentages for training and testing.

Besides, we checked that for the ZPHS01B module and $O_3$ calibration, 165 days of dataset-1 provided sufficient information to generalize the proposed models comparing with dataset-2 of 239 days. This aligns with the SFR recommended values according to (Zhu et al. (2023)). Thus, given the features and characteristics of this module, the original dataset (165 days) contains enough information to generalize the behavior of the $O_3$ sensor and their response.

As future work, we plan to expand the dataset and include complementary parameters, such as wind speed or additional metadata variables, to increase the accuracy of these models. In addition, we focus on the design of new calibration and forecasting algorithms for the different sensors embedded in the low-cost ZPHS01B module in order to improve AQ monitoring resolution.

*Data availability.* Please feel free to contact to the authors about the datasets at http://www.uv.es/eco4rupa/dataset.html

*Author contributions.* G.M.F and S.F.C contributed equally to air quality gathering process and calibration techniques, as well as coding and manuscript writing. E.M.A prepared the dataset. J.J.P.S prepared the hardware infrastructure. S.F.C and J.S.G managed the funding and external collaborations.

*Competing interests.* No competing interests are present.

*Acknowledgements.* This paper is partially funded by the Grant PID2021- 126823OB- I00 MCIN funded by MCIN/ AEI/ 10.13039/ 501100011033 and by the European Union Next-Generation EU/ PRTR; by the Generalitat Valenciana with grant references CIAICO/ 2022/ 179, CIACIF/ 2023/ 416 and CIAEST/ 2022/ 64 as well as the Spanish Ministry of Education in the call for Senior Professors and Researchers to stay in foreign centers for the grant with reference PRX23/00589.

We are grateful with the Generalitat Valenciana and its AQ monitoring networm, in particular with Rafael Orts Bargues from the Atmospheric Protection Service.

## Appendix A: Spectral analysis for $O_3$ low-cost readings from ZPHS01B module

To characterize the measurements of $O_3$, we carry out a Discrete Fourier Transform (DFT) analysis, to see the changing patterns. The DFT is a mathematical technique that transforms a discrete signal from the time domain to the frequency domain. Figure A1 shows the peaks obtained from the $O_3$ signal. There are two main peaks and their harmonics. The first peak appears in the frequency $f = 0.00025 \frac{1}{hour}$ which corresponds to a period of 4000 hours, 5.56 months, that is the total duration of data-set. The second peak indicates and reveals a relevant frequency component at $f = 0.04182251 \frac{1}{hour}$, which represents a

period of 23.91 hours (approximately 1 day). Thus, there is an $O_3$ pattern that it is repeated every day, as it could be expected in a city, based on how it is generated from road traffic by combustion engines as discussed in Section 1.

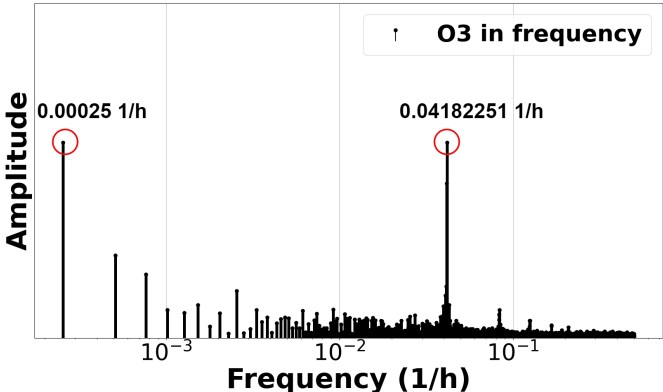

**Figure A1.** DFT of $O_3$ readings from LCS

## Appendix B: Results of models' convergence

In this appendix we plot $R^2$ and RMSE across different iterations during the training process, with training and validation datasets. Each model uses its own reference hyperparameter for convergence. In Figure B1, we observe the fit of the model in terms of $R^2$, with a better fit with training than with validation, as expected, similarly as we can see for RMSE in Figure B2. It should be noted that the convergence process with training does not reach a perfect fit in any case, which justifies and supports initially the conclusion that there is no overfitting in the models.

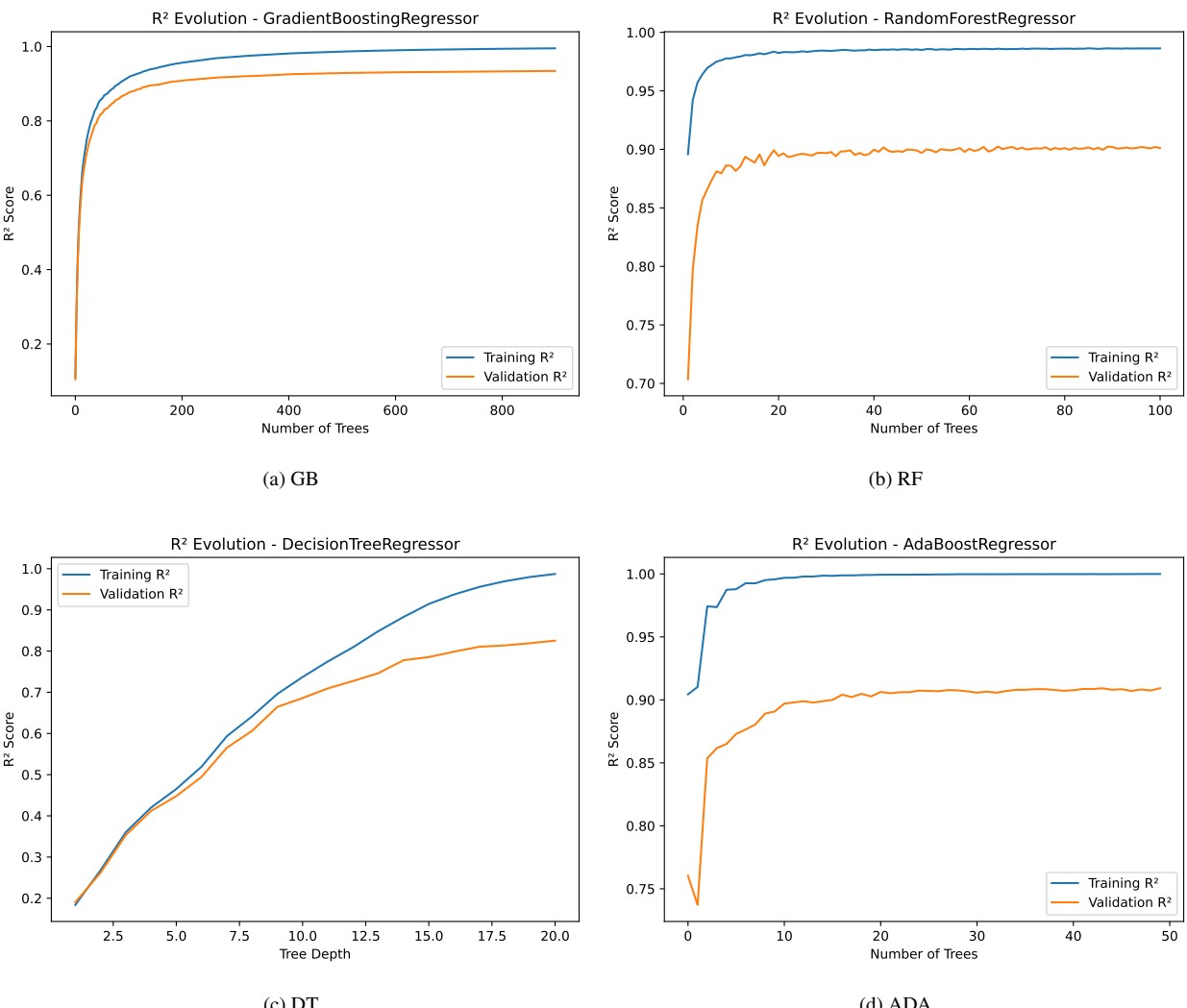

**Figure B1.** Convergence of $R^2$ across different iterations for training and validation for the models with their main reference hyperparameter.

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

405

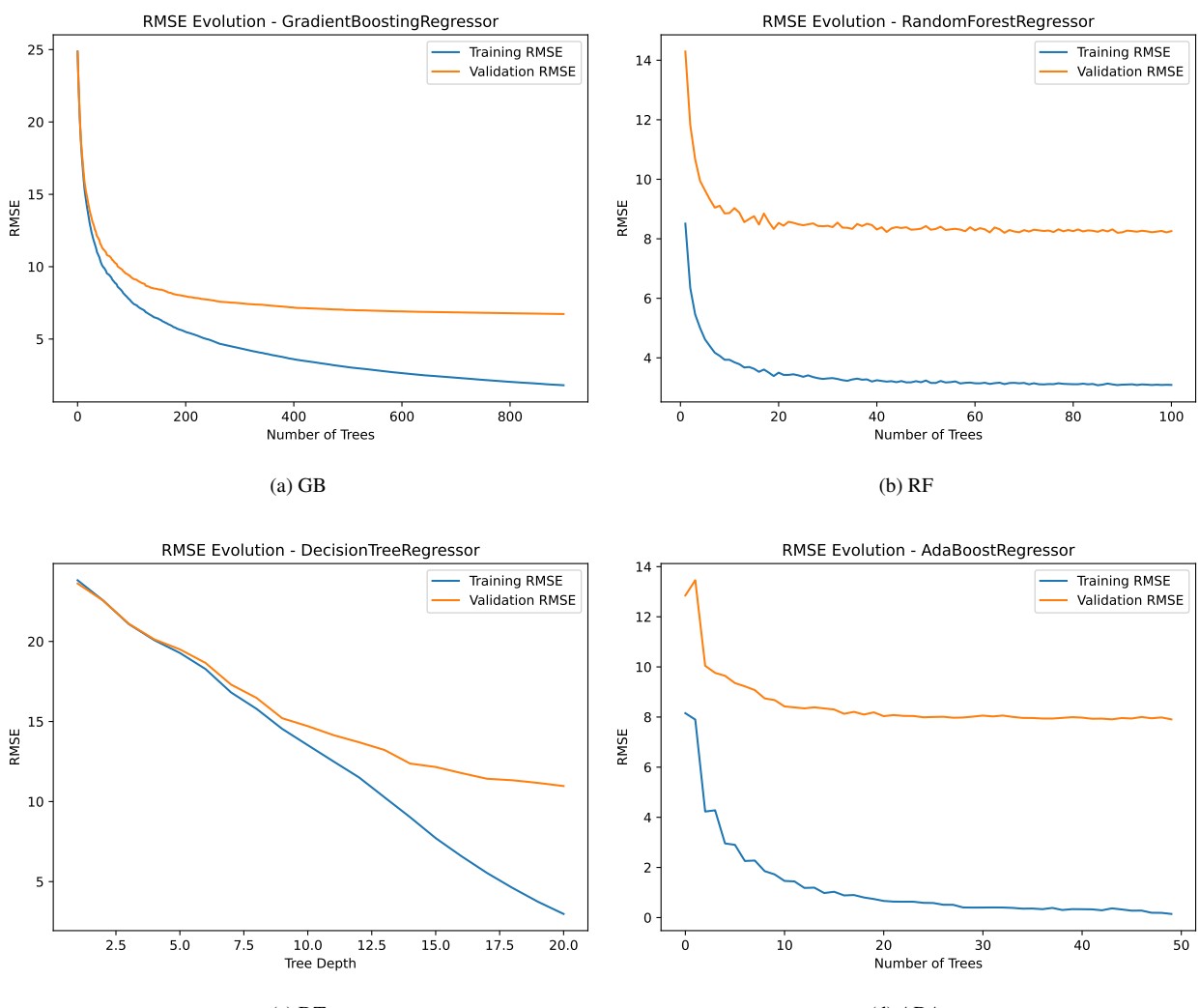

**Figure B2.** Convergence of $RMSE$ across different iterations for training and validation for the models with their main reference hyperparameter.

ment of air quality microsensors versus reference methods: The EuNetAir joint exercise, Atmospheric Environment, 147, 246–263, https://doi.org/https://doi.org/10.1016/j.atmosenv.2016.09.050, 2016.

Breeze Technologies: Air pollution – How to convert between mg/m$^3$, $\mu$g/m$^3$ and ppm, ppb, https://www.breeze-technologies.de/blog/air-pollution-how-to-convert-between-mgm3-µgm3-ppm-ppb/, 2024.

Casey, J. G., Collier-Oxandale, A., and Hannigan, M.: Performance of artificial neural networks and linear models to quantify 4 trace gas species in an oil and gas production region with low-cost sensors, Sensors and Actuators B: Chemical, 283, 504–514, https://doi.org/https://doi.org/10.1016/j.snb.2018.12.049, 2019.

Cavaliere, A., Brilli, L., Andreini, B. P., Carotenuto, F., Gioli, B., Giordano, T., Stefanelli, M., Vagnoli, C., Zaldei, A., and Gualtieri, G.: Development of low-cost air quality stations for next-generation monitoring networks: calibration and validation of $NO_2$ and $O_3$ sensors, Atmospheric Measurement Techniques, 16, 4723–4740, https://doi.org/10.5194/amt-16-4723-2023, 2023.

Corp, Z. W. E. T.: Ozone detection module ZE27-03, https://www.winsen-sensor.com/d/files/manual/ze27-o3.pdf, [Accessed 27/11/2024], 2024.

Coto-Fuentes, H., Valdés-Perezgasga, F., Guevara-Amatón, K., Limones-Ríos, K., and Calderón-Ibarra, C.: Integración de estaciones KNARIO con un sistema de información geográfico para el monitoreo de la calidad del aire en la zona metropolitana de La Laguna, Revista Ciencia, Ingeniería y Desarrollo, 1, 109–114, 2022.

DecentLab, Ltd.: Air quality sensor DL-LP8P, https://www.catsensors.com/media/Decentlab/Productos/Decentlab-DL-LP8P-datasheet.pdf, accessed: 27/11/2024, 2024.

Directive 2008/50/EC: of the European Parliament and of the Councils of 21 May 2009 on ambient air quality and cleaner air for Europe., Official Journal of the European Communities, L 152, 1–44, 2008.

Esposito, E., De Vito, S., Salvato, M., Bright, V., Jones, R., and Popoola, O.: Dynamic neural network architectures for on field stochastic calibration of indicative low cost air quality sensing systems, Sensors and Actuators B: Chemical, 231, 701–713, https://doi.org/https://doi.org/10.1016/j.snb.2016.03.038, 2016.

Felici-Castell, S., Segura-Garcia, J., Perez-Solano, J. J., Fayos-Jordan, R., Soriano-Asensi, A., and Alcaraz-Calero, J. M.: AI-IoT Low-Cost Pollution-Monitoring Sensor Network to Assist Citizens with Respiratory Problems, Sensors, 23, https://doi.org/10.3390/s23239585, 2023.

Garcia, M. A., Villanueva, J., Pardo, N., Perez, I. A., and Sanchez, M. L.: Analysis of ozone concentrations between 2002–2020 in urban air in Northern Spain, Atmosphere, 12, 1495, 2021.

García, M. R., Spinazzè, A., Branco, P. T., Borghi, F., Villena, G., Cattaneo, A., Gilio, A. D., Mihucz, V. G., Álvarez, E. G., Lopes, S. I., Bergmans, B., Orłowski, C., Karatzas, K., Marques, G., Saffell, J., and Sousa, S. I.: Review of low-cost sensors for indoor air quality: Features and applications, Applied Spectroscopy Reviews, 57, 747–779, https://doi.org/10.1080/05704928.2022.2085734, 2022.

Generalitat Valenciana: Xarxa Valenciana de Vigilància i Control de la Contaminació Atmosfèrica, Estació de Bulevar Sud, https://rvvcca.gva.es/estatico/46250050, accessed: 27/3/2025, 2025a.

Generalitat Valenciana: Xarxa Valenciana de Vigilància i Control de la Contaminació Atmosfèrica, Estació de Moli del Sol, https://rvvcca.gva.es/estatico/46250048, accessed: 27/3/2025, 2025b.

H. Adair-Rohani: Air pollution responsible for 6.7 million deaths every year, https://www.who.int/teams/environment-climate-change-and-health/air-quality-and-health/health-impacts/types-of-pollutants, accessed: 27/11/2024, 2024.

Johnson, N. E., Bonczak, B., and Kontokosta, C. E.: Using a gradient boosting model to improve the performance of low-cost aerosol monitors in a dense, heterogeneous urban environment, Atmospheric Environment, 184, 9–16, https://doi.org/https://doi.org/10.1016/j.atmosenv.2018.04.019, 2018.

Karagulian, F., Barbiere, M., Kotsev, A., Spinelle, L., Gerboles, M., Lagler, F., Redon, N., Crunaire, S., and Borowiak, A.: Review of the Performance of Low-Cost Sensors for Air Quality Monitoring, Atmosphere, 10, https://doi.org/10.3390/atmos10090506, 2019.

Kennedy, Z., Huber, D., Xie, H. R., Sohl, J. E., Page, J., and Dowell, W.: Miniature Multi-Sensor Array (mini-MSA) for Ground-to-Stratosphere Air Measurement, Phase II, Mechanical Engineering Commons, https://digitalcommons.usu.edu/cgi/viewcontent.cgi?article=1600&context=spacegrant, 2021.

Malings, C., Tanzer, R., Hauryliuk, A., Kumar, S. P. N., Zimmerman, N., Kara, L. B., Presto, A. A., and Subramanian, R.: Development of a general calibration model and long-term performance evaluation of low-cost sensors for air pollutant gas monitoring, Atmospheric Measurement Techniques, 12, 903–920, https://doi.org/10.5194/amt-12-903-2019, 2019.

Manisalidis, I., Stavropoulou, E., Stavropoulos, A., and Bezirtzoglou, E.: Environmental and Health Impacts of Air Pollution: A Review, Frontiers in Public Health, 8, https://doi.org/10.3389/fpubh.2020.00014, 2020.

Meneses-Albala, E., Montalban-Faet, G., Felici-Castell, S., Perez-Solano, J. J., and Fayos-Jordan, R.: Assessment of a Multisensor ZPHS01B-Based Low-Cost Air Quality Monitoring System: Case Study, Electronics, 14, https://doi.org/10.3390/electronics14081531, 2025.

Nova Fitness Co., Ltd.: Air quality sensor SDS011, https://cdn-reichelt.de/documents/datenblatt/X200/SDS011-DATASHEET.pdf, accessed: 27/11/2024, 2024.

Obregon, J. and Jung, J.-Y.: Chapter 4 - Explanation of ensemble models, in: Human-Centered Artificial Intelligence, edited by Nam, C. S., Jung, J.-Y., and Lee, S., pp. 51–72, Academic Press, ISBN 978-0-323-85648-5, https://doi.org/https://doi.org/10.1016/B978-0-323-85648-5.00011-6, 2022.

Okafor, N. U., Alghorani, Y., and Delaney, D. T.: Improving Data Quality of Low-cost IoT Sensors in Environmental Monitoring Networks Using Data Fusion and Machine Learning Approach, ICT Express, 6, 220–228, https://doi.org/https://doi.org/10.1016/j.icte.2020.06.004, 2020.

Organization, W. H. et al.: Air Quality Guidelines-Update 2021, Copenhagen, Denmark: WHO Regional Office for Europe, 2021.

Pedregosa, F., Varoquaux, G., Gramfort, A., Michel, V., Thirion, B., Grisel, O., Blondel, M., Prettenhofer, P., Weiss, R., Dubourg, V., Vanderplas, J., Passos, A., Cournapeau, D., Brucher, M., Perrot, M., and Duchesnay, E.: Scikit-learn: Machine Learning in Python, Journal of Machine Learning Research, 12, 2825–2830, 2011.

Seinfeld, J. H. and Pandis, S. N.: Atmospheric chemistry and physics: from air pollution to climate change, John Wiley & Sons, 2016.

Sensit: anatrac.com, https://www.anatrac.com/wp-content/uploads/2021/04/sensit-ramp-brochure.pdf, [Accessed 27-11-2024], 2024.

SGX, SensorTech: Air quality sensor MiCS-6814, https://www.sgxsensortech.com/content/uploads/2015/02/1143_Datasheet-MiCS-6814-rev-8.pdf, accessed: 21/11/2024, 2024.

Shinyei: PPD42 sensor by Shinyei Tech. Co., https://www.shinyei.co.jp/stc/eng/products/optical/ppd42nj.html, [Accessed27/11/2024], 2024.

Vaheed, S., Nayak, P., Rajput, P. S., Snehit, T. U., Kiran, Y. S., and Kumar, L.: Building IoT-Assisted Indoor Air Quality Pollution Monitoring System, in: 2022 7th International Conference on Communication and Electronics Systems (ICCES), pp. 484–489, https://doi.org/10.1109/ICCES54183.2022.9835822, 2022.

Van Poppel, M., Schneider, P., Peters, J., Yatkin, S., Gerboles, M., Matheeussen, C., Bartonova, A., Davila, S., Signorini, M., Vogt, M., Dauge, F., Skaar, J., and Haugen, R.: SensEURCity: A multi-city air quality dataset collected for 2020/2021 using open low-cost sensor systems, Scientific Data, 10, https://doi.org/10.1038/s41597-023-02135-w, 2023.

Wang, G., Yu, C., Guo, K., Guo, H., and Wang, Y.: Research of low-cost air quality monitoring models with different machine learning algorithms, Atmospheric Measurement Techniques, 17, 181–196, https://doi.org/10.5194/amt-17-181-2024, 2024.

Wang, R., Li, Q., Yu, H., Chen, Z., Zhang, Y., Zhang, L., Cui, H., and Zhang, K.: A Category-Based Calibration Approach With Fault Tolerance for Air Monitoring Sensors, IEEE Sensors Journal, 20, 10 756–10 765, https://doi.org/10.1109/JSEN.2020.2994645, 2020.

Zhengzhou Winsen Electronics Technology Co., L.: Multi-in-One Sensor Module (Model: ZPHS01B) Manual, https://www.winsen-sensor.com/d/files/zphs01b-english-version1_1-20200713.pdf, [Accessed 27/11/2024], 2024.

490     Zhu, J.-J., Yang, M., and Ren, Z. J.: Machine Learning in Environmental Research: Common Pitfalls and Best Practices, Environmental Science & Technology, 57, 17 671–17 689, https://doi.org/10.1021/acs.est.3c00026, pMID: 37384597, 2023.

Zimmerman, N., Presto, A. A., Kumar, S. P. N., Gu, J., Hauryliuk, A., Robinson, E. S., Robinson, A. L., and Subramanian, R.: A machine learning calibration model using random forests to improve sensor performance for lower-cost air quality monitoring, Atmospheric Measurement Techniques, 11, 291–313, https://doi.org/10.5194/amt-11-291-2018, 2018.