# Peer review of "Improving Raw Readings from Low-Cost Ozone Sensors Using Artificial Intelligence for Air Quality Monitoring"

_Atmospheric Measurement Techniques, 2024_

## Author Comment (AC1)

REFERENCE: amt-2024-127

**Title: "***Improving Raw Readings from Low-Cost Ozone Sensors Using Artificial Intelligence for Air Quality Monitoring***"**

**Authors:** *Guillem Montalban-Faet, Eric Meneses-Albala, Santiago Felici-Castell, Juan J. Perez-Solano and Jaume Segura-Garcia*

Departament de Informàtica, ETSE, Universitat de València, Avd. de la Universidad S/N, 46100 Burjassot, (Valencia), Spain

Dear editor and reviewer,

Thank you for giving us the opportunity to address the comments provided by the anonymous reviewers. We have made every effort to respond thoroughly to their feedback. Attached is a response letter with our responses highlighted **in blue**. The revised manuscript also uses **blue text** to indicate the changes made. In some answers, this **blue text** is highlighted if there is more than one answer.

We would also like to express our gratitude to the anonymous reviewers for their valuable comments and suggestions. We appreciate the time and effort they have invested in improving our work. We firmly believe that this manuscript is now suitable for publication and an excellent contribution to share with the broader research community.

**Reviewer's comments (Anonymous Referee #1, 31 Oct 2024)**

This paper requires major edits to be considered for publication. The introduction and related works sections are extremely weak and do not set up a solid foundation for the work the authors hope to achieve with their gradient boosted calibration of low-cost ozone sensors. The many figures and tables regarding feature selection are not well explained. The ML model generation and final model outputs, especially the gradient boosted model, seem sound, but the authors neglect to include the results of the testing data, which is a better indicator of whether the models are overfitting and better demonstrates how these models would perform in the field as compared to the training statistics, which are the focus of the article. The grammar throughout needs improvement, and there are many instances where subscript is needed (including in figures). Overall, additional literature review and context will lay a stronger foundation for the model building, and careful revision of which figures and tables are really necessary along with added information on the training dataset of the model (which speaks to overfitting and real-world applicability) will greatly improve the paper.

**Response 1:** First of all, we would like to sincerely thank you for your thoughtful review and comments, which have greatly contributed to improving our work.

In the following sections, we will address all your comments, queries, and suggestions.

The introduction section does not provide sufficient context. First, the authors list a few vague sentences about air quality in general. For example, line 15: "exceeds the limit values of the recommended safety guidelines"– what guidelines? Limits for what pollutants?

**Response 2**: We have improved this paragraph and placed the Air Quality Guideline (AQG) reference next to this sentence. Also, we have included the information from this guideline related to ground-level ozone in the text, with a target of 100 μg/m$^3$ during 8 hours in average. These changes are introduced in the second and third paragraphs of the new version, as follows:

AQ has a direct impact on both human health and the environment (Manisalidis et al. (2020)). According to World Health

15   Organization (WHO) (H. Adair-Rohani (2024)), 99% of the world's population breathes air that exceeds the limit values of the recommended safety Air Quality Guideline (AQG) (Organization et al. (2021)). This guideline specifies recommended levels for these pollutants for both short-term and long-term exposure. It is regularly reviewed and updated to incorporate the latest scientific evidence on the health effects of air pollution. This helps governments and authorities establish and implement policies to protect human health from the adverse effects of air pollution.

20   Among these pollutants, we focus on O3, a highly oxidizing gaseous pollutant, that has very reactive properties and is harmful at high levels. **Notice that in this AQG with regard to O3, the target is to achieve a concentration of 100** $\mu g/m^3$ **measured on average of daily maximum 8 hours.** **Continued exposure to levels above those recommended by this AQG**

For your information, the whole table from this AQG (not included in the manuscript) is:

| Pollutant | Averaging time | Interim target | | | | AQG level |
|---|---|---|---|---|---|---|
| | | 1 | 2 | 3 | 4 | |
| PM$_{2.5}$, µg/m³ | Annual | 35 | 25 | 15 | 10 | 5 |
| | 24-hour* | 75 | 50 | 37.5 | 25 | 15 |
| PM$_{10}$, µg/m³ | Annual | 70 | 50 | 30 | 20 | 15 |
| | 24-hour* | 150 | 100 | 75 | 50 | 45 |
| O$_3$, µg/m³ | Peak season[b] | 100 | 70 | – | – | 60 |
| | 8-hour* | 160 | 120 | – | – | 100 |
| NO$_2$, µg/m³ | Annual | 40 | 30 | 20 | – | 10 |
| | 24-hour* | 120 | 50 | – | – | 25 |
| SO$_2$, µg/m³ | 24-hour* | 125 | 50 | – | – | 40 |
| CO, mg/m³ | 24-hour* | 7 | – | – | – | 4 |

* 99th percentile (i.e. 3–4 exceedance days per year).
[b] Average of daily maximum 8-hour mean O$_3$ concentration in the six consecutive months with the highest six-month running-average O$_3$ concentration.

A brief description of how ozone is formed is given, but no specifics on the region of interest or what these health effects and consequences.

**Response 3**: To explain the effects of ozone on our health, we have included the following explanation next to the O3 details, in the third paragraph of the new version (lines 22-26 of the revised version):

measured on average of daily maximum 8 hours. Continued exposure to levels above those recommended by this AQG may lead to respiratory irritation, lung inflammation, aggravation of respiratory diseases such as asthma or bronchitis, cell damage and may have associated effects on the cardiovascular system. Those at the highest risk include children,

25 older adults, people with respiratory or heart conditions, and individuals who spend significant time outdoors (Garcia et al. (2021)).

This information is extracted from the study by Garcia, M. A., Villanueva, J., Pardo, N., Perez, I. A., and Sanchez, M. L.: "Analysis of ozone concentrations between 2002-2020 in urban air in Northern Spain", Atmosphere, 12, 1495, 2021.

The authors mention that low-cost sensors have lower accuracy – why? What are the issues surrounding them?

**Response 4**: Thank you for your comment. In order to clarify this issue, we have improved the explanation about the accuracy of low-cost sensors, based on the following criteria: a) sensor technology, b) calibration, c) environmental sensitivity, d) limited range and sensitivity, e) materials and build quality, f) sensor cross-sensitivity, g) maintenance and

lifespan. These details about low-cost sensors are also found in several references that have been included in the new version of the manuscript as follows (lines 32-39):

> ing to Annex III-B of the European (Directive 2008/50/EC (2008)). And Low-Cost Sensor (LCS) are becoming increasingly important, an interesting alternative, but they do not have good accuracy (Borrego et al. (2016)) in comparison with the official equipment, due to limitations in their sensing technology, lack of frequent calibration, sensitivity to environmental
> 35    factors, cross-sensitive issues, use of less durable materials and the absence of rigorous certification processes. While official equipment uses advanced technologies and is subject to strict standards of accuracy and reliability, LCS are designed to offer basic monitoring at a low price, which involves sacrifices in accuracy and durability. So, in this context it is a challenge to estimate the official measurements from these LCS with a reduced error (García et al. (2022); Borrego et al. (2016)).

One other machine learning-enabled calibration effort is mentioned in this section, with no information on HOW machine learning actually improves this. This feels out of place here and the same information is listed again in section 2, so I would suggest removing it here and expanding on it in section 2. More exploration of other machine learning based calibration algorithms beyond the ZPHS01B-specific ones referenced later on would strengthen the paper as ML-based calibration is common practice in the field. The outline in line 36 is unnecessary.

**Response 5**: Thank you for your comments. We have improved the wording and retained Zimmerman's reference and details only in the related work section. In the introduction, we have instead included a more general reference regarding best practices in machine learning methods for environmental research, based on a review of over 148 highly cited research papers. Additionally, we have introduced the machine learning algorithms used in this paper, along with their abbreviations. For better readability and clarity, we have left the outline with a better wording."

Thus, all these changes we have been introduced in the last part of this introduction section as follows (lines 40-54):

> 40    Artificial Intelligence (AI) techniques are valuable for environmental research due to their capacity to process large datasets and identify patterns that enhance system explainability and clarify the behavior of these AQ parameters (Zhu et al. (2023)).
>
>      In this paper, we show that Machine Learning (ML) models, in particular ensemble models, can correct the raw readings from LCS by taking into account additional environmental information, such as Temperature (Temp), Relative
> 45    Humidity (RH), as well as other pollutants, being able to use these sensors to extend the resolution of air quality monitoring networks at low cost, but assuming a small error. This is our main objective. We propose and compare different techniques, reducing the estimation error up to 94.05% based on Mean Absolute Error (MAE) measurements, with a Mean Relative Error (MRE) of 7.21%, achieving the best results with the Gradient Boosting (GB) algorithm and outperforming the related work, using sensors approximately 10 times less expensive. We also carry out the calibration process using Random
> 50    Forest (RF), Adaptive Boosting (ADA) and Decision Tree (DT) models.
>
>      The rest of the paper is structured as follows. Section 2 introduces the related work. Section 3 explains the experimental work carried out for the deployment of LCS and shows the data processing, as well as the use of ML algorithms for the O3 calibration of these LCS. The results are shown in Section 4 and finally, the conclusions and future work are presented in Section 5.

In table 1, rather than listing "low", "mid-low", etc. and then defining it in the text, it would be easier for the reader if the cost was just listed in the table.

**Response 6**: We have included the price range information in the caption of this Table 1 and in the text in the related work section, as follows (lines 59-65):

| Module | Sensors | Relative cost |
|---|---|---|
| SDSO11 (Nova Fitness Co., Ltd. (2024)) | Temp, RH, PM, PA | Low |
| DL-LP8P (DecentLab, Ltd. (2024)) | Temp, RH, CO2, PA | Low |
| MiCS-6814 (SGX, SensorTech (2024)) | CO, NO2, C2H5OH, NH3, CH4 | Low |
| ZPHS01B (Zhengzhou Winsen Electronics Technology Co. (2024)) | Temp, RH, PM1-10, CO, CO2, O3, NO2, TVOC | Mid-Low |
| Sensit RAMP (Sensit (2024)) | PM2.5, CO, CO2, NO, NO2, O3 | High |
| AirSensEUR (Van Poppel et al. (2023)) | NO, NO2, O3, CO, PM2.5, PM10, PM1, CO2 | Mid-High |

**Table 1.** AQ Sensor modules with cost estimate: Low (less than 10$), Mid-Low (100-200$), Mid-High (600-1000$) and High ($\approx$<4000$).

Since in AQ different pollutants are considered and each sensor measures only one, we will analyze sensor modules
60  that embed some of these LCS. A list of these sensor modules with a cost estimate is given in Table 1. The selection criteria of these modules is determined by the related work, selecting those modules which have been considered under a similar studies as the proposed here. We must stress that these modules have different costs due to their quality, order quantity, country, etc. that we can classify in: Low (less than 10$), Mid-Low (100-200$), Mid-High (600-1000$) and High ($\approx$<4000$). A larger selection and comparison of these LCS modules are given in (García et al. (2022)) and (Borrego
65  et al. (2016)).

Better distinction is needed between what is an individual sensor vs. what is a complete package. The table is titled systems and/or modules, but the text does not explain what the distinction between a system or module is.

**Response 7**: Thank you for this comment. In this paper, there are no differences between modules and systems since both refer to multi-sensor platforms. We have unified these words as modules, as you can see in the previous response.

Why were these chosen for this table? Without any explanation as to why these are here it seems random.

**Response 8**: The selection criteria of these modules is determined by the related work, selecting those modules which have been considered under similar studies as the proposed in this manuscript. These details have been introduced in the new version of the manuscript as you can see in the same paragraph depicted in Response 6. The highlighted part contains this explanation as seen below (lines 60-62):

Since in AQ different pollutants are considered and each sensor measures only one, we will analyze sensor modules
60  that embed some of these LCS. A list of these sensor modules with a cost estimate is given in Table 1. The selection criteria of these modules is determined by the related work, selecting those modules which have been considered under a similar studies as the proposed here. We must stress that these modules have different costs due to their quality, order quantity, country, etc. that we can classify in: Low (less than 10$), Mid-Low (100-200$), Mid-High (600-1000$) and High ($\approx$<4000$). A larger selection and comparison of these LCS modules are given in (García et al. (2022)) and (Borrego
65  et al. (2016)).

There is also no explanation as to why some are more expensive than others – are some better performing?

**Response 9**: We have added additional information to justify this point regarding differences in their prices. Note that even within this low-cost range, there are variations in the quality of LCS production, such as the materials used and sensor calibration, which affect accuracy and durability.

These details have been introduced in the new version as follows (lines 66-74):

> Note that LCS are designed for basic monitoring at a low cost, which compromises accuracy and durability. In this list, there are several types of LCS. Optical type sensors, such as SDSO11 (Nova Fitness Co., Ltd. (2024)) and DL-LP8P (DecentLab, Ltd. (2024)), that measure the amount of light absorbed by a given gas. Metal-oxide sensors, such as SGX, SensorTech (2024) that measure the change in electrical conductivity on a semiconductor due to the presence
> 70  of certain gases. Usually this type of sensors are the cheapest and are particularly susceptible to cross sensitivities. And electrochemical sensors that have higher selectivity, good for measuring specific gases, but they are more expensive. Among these, Sensit RAMP (Sensit (2024)) and AirSensEUR (Van Poppel et al. (2023)) use this type of sensors. Finally, the ZPHS01B module (Zhengzhou Winsen Electronics Technology Co. (2024)) integrates optical, metal-oxide and electrochemical sensors and it is a Mid-Low price module with the best *price/sensor* ratio.

In line 48, "it is necessary to use modules embedding as many AQ LCS as possible." - why is it necessary?

**Response 10**: This is necessary for us because gases are often correlated with other gases (cross-sensitivity issues) and with factors not directly related to air quality, such as temperature and relative humidity. To clarify these points, we have improved the wording and included references that support these arguments.

These details are reflected in the revised version of the manuscript as follows (lines 75-77):

> 75  Since one of the key points to improve the accuracy of these LCS is the use of marginal information (such Temp, RH as well as other AQ pollutants), **exploited using AI techniques (Karagulian et al. (2019); Esposito et al. (2016)) as mentioned before**, it is necessary to use multi-gas modules embedding as many AQ LCS as possible.

In line 49, it is stated that one of these "is the best solution at the time of writing". If this means the best choice for the author's specific set of needs and wants, this needs to be clearly stated. It reads as an opinion stated as universal fact. Table 2 does not summarize 4 distinct concentration levels as stated in the text.

**Response 11**: Following the previous response, in the manuscript we have included a better explanation of this statement and explained better the output of TVOC sensors with 4 levels from this ZPHS01B module.

All this information is included in the next paragraph (lines 78-87):

> Thus, among the different low-cost alternatives and taking into account the number of sensors and the *price/sensor* ratio, the ZPHS01B (Zhengzhou Winsen Electronics Technology Co. (2024)) is the AQ sensor module that best meets the

80    **the needs and objectives of this study at the time of writing**, since it embeds 9 different sensors: Temp(°C), RH (%), as well as CO, CO2, NO2, O3 that are measured in Parts Per Million (ppm), formaldehyde (CH2O) that is measured in mg/$m^3$, PM measured in $\mu g/m^3$ and TVOC that is measured using **4 levels according to its concentration (0-very low, 1-low, 2-intermediate and 3-high)**. Table 2 summarizes all this information. Notice that the O3 sensor used in this module is the electrochemical ZE27-O3 (Corp (2024)) that measures within the range 0-10 ppm with a resolution of 0.01 ppm. It operates

85    with an accuracy of ±0.1 ppm when the concentration is ≤1 ppm and ±20% when the concentration is above 1 ppm. Also, notice that the PM readings in this module are given for 2.5 (fine particles with a diameter of 2.5 $\mu m$), and PM1 and PM10 are estimated from the PM2.5 readings.

In line 66, the authors state "The calibration process of these LCS is a challenge, where ML and Deep Learning (DL) models can be used." The authors have not given any information on why calibrating low-cost sensors is challenging. The introduction should include more background information on what these challenges are.

**Response 12**: We have improved these issues and motivated better these challenges regarding the calibration of low-cost sensors as shown before in Response 4.

This explanation is given in the new version as follows in Section I (lines 32-39):

ing to Annex III-B of the European (Directive 2008/50/EC (2008)). And Low-Cost Sensor (LCS) are becoming increasingly important, an interesting alternative, but they do not have good accuracy (Borrego et al. (2016)) **in comparison with the official equipment, due to limitations in their sensing technology, lack of frequent calibration, sensitivity to environmental**

35    **factors, cross-sensitive issues, use of less durable materials and the absence of rigorous certification processes. While official equipment uses advanced technologies and is subject to strict standards of accuracy and reliability, LCS are designed to offer basic monitoring at a low price, which involves sacrifices in accuracy and durability. So, in this context it is a challenge to estimate the official measurements from these LCS with a reduced error (García et al. (2022); Borrego et al. (2016))**.

Also, in the related work these challenges are stressed again as shown at the beginning of the following paragraph included in Section 2, line 97 from the new version:

The calibration process of these LCS is a challenge as mentioned before, where ML and Deep Learning (DL) models can be used. In (Zimmerman et al. (2018)), the authors show calibration models (using 16 weeks data) to improve sensor performance,

There are numerous other papers using gradient boosting to calibrate low-cost sensors, yet there is not even one cited in this 'related work' section.

**Response 13**: Thank you for your remark. In the related work section of the revised manuscript, we have included new references that use the gradient boosting algorithm. In total, there are eight references on machine learning algorithms, and we have selected three for direct comparison, as shown in Table 14, 'Comparison with Similar Related Work.' Additionally, we have improved the paragraph discussing the use of ML algorithms and gradient boosting, as mentioned in the previous response.

This update is reflected in Section 2, starting at line 97, with two additional references added, as follows (lines 100-104):

100 system), they achieve an MRE of 15% for O3. **In the study performed by (Johnson et al. (2018)), the calibration of an aerosol sensor for PM2.5 is carried out by comparing simple linear regression models with GB using the PPD42 PM sensor (Shinyei (2024)). The study concludes that gradient boosting performed better and significantly improved the performance of the sensors, reaching a coefficient of determination ($R^2$) of up to 0.76.** In (Casey et al. (2019)), the

The final two paragraphs of section 2 are both non sequiturs. The authors do not mention data preprocessing, analysis or interpretability at all up to this point – this paragraph would only make sense if information on how others have handled these aspects of the data were included in the literature review of other ML calibration techniques.

**Response 14**: We appreciate your comment and have enhanced the revised manuscript by including detailed information about this data preprocessing in Section 2. We have introduced these concepts, which are later used in the analysis of the algorithms in subsequent sections.

Additionally, in Section 2, we have discussed how related work has addressed these issues, focusing on Feature Importance Analysis (FIA), Principal Component Analysis (PCA), Feature Selection (FS), and Hyperparameter Optimization (HPO)."

This enhanced paragraphs in Section 2 is as follows (lines 111-131):

In this context, when using AI techniques on environmental research, it is important to follow the recommendations given by ( Zhu et al. (2023)) based on a review of more than 148 highly cited research papers. In this reference, it is highlighted that data preprocessing, analysis and interpretability are often overlooked, such as Feature Importance Analysis (FIA), Principal Component Analysis (PCA) and Feature Selection (FS). **In addition, it is said that the process of optimizing algorithms**

115 **through the selection of their hyperparameters (Hyperparameter Optimization (HPO)) is neglected in most of the environmental research studies considered. For instance, in (Johnson et al. (2018)), better results are obtained with GB, but HPO is not performed in the model, which could allow further improvements of the results. Both (Malings et al. (2019)) and (Borrego et al. (2016)) take into account some aspects related to the data analysis focused on the optimization of the problem, but they do not carry out a HPO. In (Esposito et al. (2016)), the authors carry out a kind**

120 **of simple HPO, based on raw tests of different architectures and modifying hyperparameters, such as the number of hidden layers of the model, tapped delay length and feedback delay line length, concluding that a dynamic approach to these parameters improves the results with respect to a static approach without changing the value of these parameters.**

**Regarding the selection of parameters, in (Johnson et al. (2018)), the authors does not perform an analysis using techniques such as the aforementioned FIA and FS, but a sensitivity analysis using different meteorological variables**

125 **(such as Temp and RH), determining that it is useful information for GB. In (Malings et al. (2019)), the quantification of the importance of the model variables is mentioned as a mean to understand which information is useful, concluding that for RF, to add additional information apart from AQ measurements, such as Temp and RH are very helpful. In (Borrego et al. (2016)) and (Esposito et al. (2016)), the authors do not include a specific analysis of the relative importance of different variables or features. However, a good example of FS is depicted in (Okafor et al. (2020)), where**

130 **it is shown that identifying the environmental factors affecting LCS is crucial for improving data quality using data fusion and ML. These factors are then incorporated into the development of the calibration model.**

On line 80, the authors write, "In conclusion, we see that to increase the AQ monitoring resolution at a city scale, LCS are required." This has nothing to do with the related works in this section, where different machine learning algorithms and their previous performances are listed.

**Response 15**: We have improved this explanation as it concludes both Sections 1 and 2. To emphasize the main goal of this manuscript, and drawing on the information provided by Directive 2008/50/EC (2008) and the reference by Zhu et al. (2023) on best practices in applying machine learning methods in environmental research, we have added the following paragraph at the end of Section 2 (lines 132-137):

> In conclusion, in order to increase the resolution of city-scale AQ monitoring according to the recommendations given by (Directive 2008/50/EC (2008)) as mentioned before, it is necessary to perform a calibration process of these LCS. In

> this scenario, we focus on the O3 calibration by using ensemble ML techniques, comparing different techniques. For
> 135  this purpose, it is necessary to carry out a thorough data treatment with a good practice criteria (Zhu et al. (2023)) including HPO, FIA and/or FS, which are usually overlooked. In this case, we achieve interesting results compared with the related work, as shown in Section 4.

Table 3 seems unnecessary since most of the data available at this station was unused, and it seems the relevant ones are already listed in the text?

**Response 16**: Right. We have deleted this table and only included in the text the information about the O3 measurements from the AQ official monitoring stations, as highlighted within the following paragraph in the new version of the manuscript (lines 146-147):

> To calibrate the O3 sensor from the ZPHS01B module, we require a dataset to train various ML models. For this purpose, we use as reference values, O3 concentration readings from the official AQ station in the Valencian AQ Monitoring Network
> 145  (VAQMN), at Bulevar Sur (Valencia, Spain) managed by Generalitat Valenciana (GVA) with latitude and longitude 39.450389 and -0.396324, respectively, as shown in Figure 1. These reference values are given in $\mu g/m^3$ periodically averaging every 10-minutes. The AQ station data is retrieved from *https://rvvcca.gva.es/estatico/46250050*. The ZPHS01B module's readings are taken at a rate of 10 samples per minute.

In section 3.2, the monitoring intervals listed on lines 104-105 are unclear. Is this 10 minute average or once every 10 minutes? The comment on line 105 "it is sufficient" is also unclear – you need to explain to the reader why without expecting them to read the entire Zhu paper.

**Response 17**: Thank you for your comment. The monitoring intervals used are the average of the original ZPHS01B module readings, taken at a rate of 10 samples per minute, one sample every 6 seconds, as depicted in line 147-148 of the new version.

Then, we have enhanced this explanation about these intervals in Section 3.2 "Analyzing the data set" as detailed next. Notice that based on the number of samples and the number of features (in practice we used 5), it is analyzed the Sample-size to Feature-size Ratio (SFR) according to the Zhu's paper.

These explanations and its improved paragraph are included as follows (lines 156-163):

**3.2 Analyzing the dataset**

The initial data collection is based on 6-second frequency samples, including 165 days (approximately five and a half months), from June $8^{th}$ 2023 till November $20^{th}$ 2023. Based on this collection, three datasets have been created by averaging data over different time monitoring intervals: 10 min., 30 min. and 1 h. with 23496, 7843 and 3922 samples
160 respectively. The lowest 10 min. interval is given by the official AQ station and 30 min and 1 h are common time base for AQ parameters. Although they are not large data-set, it is sufficient as shown in (Zhu et al. (2023)), due to the relationship (ratio) between sample size and feature size, 5 in total as seen next. This ratio is called, Sample-size to Feature-size Ratio (SFR), being recommended a SFR higher than 500. More detail is given in Section 3.3.

For any table or figure, the reader should be able to understand it based on the table or figure and its caption alone. For table 4, the meaning of the abbreviations are not defined anywhere in the figure, caption, or main text. You shouldn't make the reader guess what MAD, Diff., Stat., etc. stand for. Without any definitions, this information is not helpful to the reader. Even with definitions, it's a huge jump from this table to what's written in the text.

**Response 18**: We value your suggestion. We have included all these details in the caption of this Table (Table 3 in the new version) and in the explanation of the accompanying text as follows (lines 168-172):

**Table 3.** Summary of main statistics of the Dataset: Minimum (Min.), Maximum (Max.), Mean (Mean), Standard Deviation, Variance (Var.), Median Absolute Deviation (MAD), percentage of samples taking Different values (Diff.), Stationarity (Stat.), Seasonality (Seas.) and High correlation (High corr.)

|  | Temp | RH | PM2.5 | CO2 | NO2 | CO | CH2O | TVOC | O3 | O3ref |
|---|---|---|---|---|---|---|---|---|---|---|
| **Min** | 5.24 | 62.29 | 21.25 | 693.43 | 0.78 | 0 | 0.005 | 0 | 39.57 | 8.71 |
| **Max** | 42.26 | 118 | 83.69 | 1792.50 | 18.81 | 0.75 | 1.21 | 2.95 | 255.76 | 97.85 |
| **Mean** | 20.60 | 91.31 | 49.99 | 780.33 | 15.27 | 0.34 | 0.021 | 0.024 | 114.39 | 55.72 |
| **SD** | 5.70 | 18.12 | 18.14 | 57.16 | 5.65 | 0.28 | 0.02 | 0.13 | 67.11 | 24.83 |
| **Var.** | 32.57 | 328.41 | 329.34 | 3268.29 | 31.92 | 0.08 | 0.0006 | 0.016 | 4503.98 | 616.69 |
| **MAD** | 3.92 | 16.37 | 13.31 | 24.53 | 0.59 | 0 | 0.001 | 0 | 51.40 | 16.21 |
| **Diff.** | 99.1% | 81.9% | 87.9% | 97.5% | 50.6% | 0.2% | 81.2% | 5.8% | 75.0% | 30.3% |
| **Stat.** | not | not | not | not | not | not | yes | yes | not | not |
| **Seas.** | yes | yes | yes | yes | yes | yes | not | not | yes | yes |
| **High corr.** | yes | yes | yes | yes | not | yes | not | not | yes | yes |

Table 3 shows a summary of main statistics of the dataset. For each parameter is shown: the Minimum value (Min.), Maximum value (Max.), Mean value of all entries (Mean), Standard Deviation, Variance (Var.), Median Absolute Deviation (MAD), percentage of samples taking Different values (Diff.), Stationarity (Stat.), Seasonality (Seas.) and High correlation (High corr.) with others. Seasonality refers to recurring patterns at regular intervals, while stationarity indicates constant statistical properties over time.

Of table 4, the authors write, "From these results, it is worth mentioning that the CH2O, CO, NO2 and TVOC sensors are not very reliable in the ZPHS01B module. Also, the RH

sensor has a positive offset as we can see from the maximum value, 118%. The other sensors have a normal behaviour, although with low accuracy." There is no CH2O, CO, or TVOC data in table 4. For NO2, the only pollutant mentioned in your description of table 4 that even appears in the table, I don't know what about the random assortment of numbers and yes/no's in the table is supposed to tell me that it's 'not very reliable'. For RH, I don't see any value of 118% in the table. Are 'RH' (as written in the text) and 'Hum' (as written in the table) different? The text and the table have almost nothing in common, and neither helps me understand what you're doing with the data.

**Response 19**: Thank you for your comment. Continuing with the previous response, initially in the manuscript, we did not include these sensors (CH2O, CO, NO2 and TVOC) since after their analysis we realized that they did not seem to be working properly in the ZPHS01B module, at least under the atmospheric conditions during the creation of the data-set. However, this information about this behavior is interesting and it has been added both to this table (now Table 3 in the new version, depicted in Response 18) and in the text. Notice that from Table 3, we can see the maximum value for Relative Humidity (RH) sensor is 118%. Also, we have fixed (standardized) this notation regarding RH in this table as well as in the whole manuscript.

The revised paragraph with all these details is as follows (lines 173-178):

> From these results, it is worth mentioning that the CH2O, CO, TVOC and NO2 sensors do not seem to be working properly in the ZPHS01B module. CH2O, CO and TVOC are almost always stuck to values close to zero, seeming not
> 175 to excite at normal concentrations, with very low variability. On the other hand, the NO2 sensor appears saturated, stuck at the maximum value, 10 ppm. Thus in practice, the number of used features from Table 3 are 5, that is from the initial 9 (the reference is not included), we remove these 4 (CH2O, CO, TVOC, and NO2). Also, RH sensor has a positive offset as we can see from the maximum value, 118%.

On line 114, DFT is not defined. After reading the rest of the section, it is never explained HOW the results of Figure 3 are used in your analysis. What do those peaks and harmonies tell you, or how do they inform the way you built the model? This needs better explanation for the figure to be worth keeping.

**Response 20**: The Discrete Fourier Transform (DFT) analysis carried out is used to see the O3 changing patterns during the gathering process. Then, we observe a daily pattern driven by road traffic.

We have included the DFT definition and this explanation in the next paragraph in the new version of the manuscript (lines 179-185):

To characterize the measurements of O3, we carry out a Discrete Fourier Transform (DFT) analysis, to see the changing
patterns. **The DFT is a mathematical technique that transforms a discrete signal from the time domain to the frequency domain.** Figure 3 shows the peaks obtained from the O3 signal. There are two main peaks and their harmonics. The first peak appears in the frequency $f = 0.00025 \frac{1}{hour}$ which corresponds to a period of 4000 hours, 5.56 months, that is the total duration of data-set. The second peak indicates and reveals a relevant frequency component at $f = 0.04182251 \frac{1}{hour}$, which represents a period of 23.91 hours (approximately 1 day). Thus, there is an O3 pattern that it is repeated every day, **as it could be expected in a city, based on how it is generated from road traffic by combustion engines as discussed in Section 1.**

In the figure 4 and 5 captions, 'vs' is typically reserved for Y vs X. Your reference and sensor ozone are plotted on the same X axis; consider rewording. In my opinion, Figure 4 can be removed as Figure 5 shows the same information but in better detail.

**Response 21**: Thank you. We have fixed these captions (changing "vs" by "and") and left only one figure from these two. We have kept only the figure that gathers a whole week (Figure 4 in the new version), with better resolution.

In table 5, some of the model acronyms are not defined in the text until well after their first appearance in the tables – moving these higher in the text or defining them directly in the table will make it easier on the reader. There is again discrepancy between 'RH' in the text and 'Hum.' in the figure. Was there a cutoff number to determine which were the most important? Was this across all models, or were the results of one in particular favored? Including this information in the text will help the reader to follow how you selected the three inputs to move forward with. I think the sentence "For clarity it is not included the importance of date and ozone itself from LCS values, that complete the rest." is meant to explain why ozone isn't included in this analysis, but the sentence doesn't make sense. It might make more sense to include ozone in the analysis to demonstrate how important it is rather than ask the reader to just trust that it is.

**Response 22**: We appreciate your comment. We have improved this table (now Table 4 in the new version), including all these details, the abbreviations of the different models (Random Forest (RF), Gradient Boosting (GB), Adaptative Boost (ADA) and Decision Tree (DT)) as well as all the different features analyzed when performing the Feature Importance Analysis (FIA). Notice that these abbreviations (models' acronyms) were already introduced in Section 1, line 49, as we depicted in Response 5. Also, in the paragraph below is explained the threshold (8%) used for the selection of the different features for the models. Also, we have fixed as seen in Response 19 the RH abbreviation.

Both Table 4 and this new paragraph are as follows (lines 193-197):

**Table 4.** FIA of ozone's complementary parameters for **Random Forest (RF), Gradient Boosting (GB), Adaptive Boost (ADA) and Decision Tree (DT)**

| Model | Temp | RH | PM2.5 | CO2 | NO2 | O3ref | CO | TVOC | CH2O |
|-------|------|------|-------|-------|-------|-------|-------|-------|-------|
| RF | 0.128 | 0.103 | 0.069 | 0.222 | 0.078 | 0.269 | 0.002 | 0.003 | 0.064 |
| GB | 0.107 | 0.105 | 0.052 | 0.211 | 0.057 | 0.253 | 0.001 | 0.001 | 0.068 |
| ADA | 0.119 | 0.097 | 0.064 | 0.246 | 0.067 | 0.287 | 0.001 | 0.001 | 0.066 |
| DT | 0.115 | 0.088 | 0.070 | 0.232 | 0.061 | 0.276 | 0.001 | 0.002 | 0.061 |

Table 4 shows the normalized output of the FIA using the *scikit-learn* library (Pedregosa et al. (2011)), for the parameters complementary to O3, for each ML models used. **In order to determine the most useful parameters for the models, a threshold is established in 8% of importance. Notice that the pattern of parameters with the highest importance, is repeated for all models.** From this analysis, we conclude that Temp, RH and CO2 are the most relevant and then will be considered for the next step (FS analysis), since they show the highest values.

In Figure 6, 'CH2O' (letter O) seems to be misspelled as 'CH20' (number zero).

Many variables that were left out of the previous tables/figures are now shown here – CH2O, CO, TVOC. Had you already ruled these out? It seems that these are in the wrong order, at the very least. Tables 4 and 5, and Figures 3-6 all seem to be getting at which data to include in the model, but several of them could likely be moved to the supplement (or removed outright) pending better explanations of how these are actually used. What separate purpose does each of them serve?

**Response 23**: We fixed this, CH20 to CH2O. Also, as depicted in the previous response, we included all these features again in Table 4 (FIA) for the different models.

About the supplementary material, we did not think about it since we included everything in Section 3 (Building the dataset and using Machine Learning algorithms). All this information is used to analyze the different features (variables) and their contribution, following step by step the recommendations given by the mentioned Zhu's paper for best practices applied on machine learning methods in environmental research as mentioned in Section 2 (lines 111-132). Further explanation was also given in Responses 5, 14 and 15.

The utility of all this information provided in this Section 3 (for FIA, FS and HPO analysis) was already included and considered in the manuscript. Besides, we have improved the wording of the manuscript to clarify these issues.

However, for a better response for the reviewer we provide clarification of these issues:

Table 3 "Summary of main statistics of the data set", shown in Response 18, serves to know the dataset of the low-cost sensors.

Table 4 (FIA) shown in Response 22, is useful to see the importance of the most important features (variables) for the models. This indicates which are the features giving more information to the models.

Figure 3 (DFT analysis) is to detect changing patterns in the ozone measurements and to see a justification of O3 related to road traffic.

Figure 4 (Instantaneous O3 readings) serves to see the dataset of 03 low-cost and the O3 reference (real value).

Figure 5 (LCS readings and O3 reference correlation matrix) serves to see how the variability of a feature explains the variability of the rest, i.e. the correlation.

The correlation between features, together with their importance for the model, is relevant information when choosing a subset of parameters to train the different models.

In the paragraph beginning on line 136, the authors state, ", two of them showed better results" – which two? List this information here.

The location of tables 6 and 7 in the text doesn't make sense – you are showing the results of the models before explaining what the models are in section 3.4. I don't think showing both tables 6 and 7 is necessary.

**Response 24**: We appreciate this comment. The reviewer is correct. We had provided results from Section 4 in Section 3 in order to define the selection of features. However, since the goal of Section 3 is to analyze the data-set and to adjust the different algorithms, once we have performed it, at the beginning of Section 4, we start defining the final features selected based on [date, O3, Temp., RH], as depicted in lines 263-264. These features are then used to carry out the training process of these algorithms for the results in Section 4. Thus, this is clarified with the following paragraph (lines 263-264)

> In addition, from the analysis carried out in Section 3.3, for the feature selection, we have proceeded in this section with the ones with the best results, based on [date, O3, Temp, RH]. We see that fewer features, better results, i.e. increasing
> 265   the SFR. Then, other dimensionality reduction techniques are not required. If we add more features that are not so significant, it makes the dataset poorer.

And then for clarity according to the reviewer, we have deleted these tables from Section 3 (Tables 6 and 7 in the first version), since they are already included in Section 4.

The authors state, "Thus, if we add more features that are not so significant, it makes the dataset poorer." This is already a well-established principle in the field that does not require explicit demonstration. You've already shown in several figures and tables how you did feature selection – does this contradict the feature selection work you did earlier? Either way, there are many other papers establishing ozone sensor + temperature + humidity (and sometimes NOx) as the best model inputs for O3 (see several below). When many others have already demonstrated the same result that it's taking you 4 tables and 4 figures to describe, you can just cite those who have done it before with a brief explanation.

https://doi.org/10.3390/atmos12050645

https://doi.org/10.5194/amt-11-1937-2018

https://doi.org/10.1016/j.snb.2018.12.049

**Response 25**: We appreciate this comment. Since this is our first experience with the low-cost multisensor module (ZPHS01B) and no prior information was available, we decided to conduct comprehensive tests to extract as much information as possible without making any assumptions. We acknowledge that similar studies exist for other types of sensors; however, for this specific module, we found no accurate information and could not predict its behavior. Additionally, low-cost sensors like this often suffer from cross-sensitivity issues, which depend on various factors.

As indicated in our previous responses (17-24) and detailed in Section 3 of the manuscript, we performed a thorough, step-by-step analysis of the dataset. Following this analysis, we focused on a subset of the data in Section 4, based on [date, O3, Temp., RH], as outlined in lines 263-264, which provided the best results.

Regarding the reference provided, we believe the first one fits better in the related work section and have included it in Section 2 as follows (lines 103-107):

> performance of the sensors, reaching a coefficient of determination ($R^2$) of up to 0.76. In (Casey et al. (2019)), the authors show that Neural Networks (NN) generally outperform lineal models to quantify O3, CO, CO2, and CH4 in
> 105 ambient air, using gas sensors integrated into U-Pod air quality monitors. Besides, they highlight that NN capture the complex nonlinear interactions among multiple gas sensors, considering factors such as Temp, RH and atmospheric chemistry. In (Borrego et al. (2016)), the authors carry out a performance evaluation during two-weeks data of the calibration

In tables 8-11, the captions should indicate what the numbers in bold mean. This is stated once in the text in line 167, but the authors don't state what criteria was used to decide on the 'best option'. Was it highest R2, lowest RMSE? If needed, all of these except for that of the best performing model can be moved to the supplemental.

**Response 26**: Thank you for this observation. We have improved the wording in Section 3.4 to clarify these issues about the Hyperparameter Optimization (HPO). In the revised version, the tables that include the selected hyperparameters are Tables 5, 6, 7 and 8 for RF, GB, ADA and DT algorithms respectively. Also, it is explained the meaning of the bold hyperparameters, the best one that optimize the different models taking into $R^2$, Root Mean Square Error (RMSE) and MAE, as depicted in line 219-220. Besides, all these 4 tables include in their caption ""showing in bold the combination that gives the best results in terms of $R^2$, RMSE and MAE".

Thus, this information is already included in Section 3.4, as follows:

Next, we discuss the different supervised ML algorithms used and the selection of the different hyperparameters taking into account the best results of $R^2$, Root Mean Square Error (RMSE) and MAE.

**3.4.1 Random Forest (RF)**

Table 5. RF hyperparameters evaluated on *GridSearch* showing in **bold** the combination that gives the best results in terms of $R^2$, RMSE and MAE.

| No. of estimators | Max. depth | Max. features |
|---|---|---|
| 50, **100**, 250, 500, 900 | 2, 5, 7, **None** | sqrt, log2, 0.1, 0.3, 0.5, **1.0** |

Notice that this detail is included also for GB, ADA and DT algorithms.

In line 202, it is stated that the 90-10 test-train split worked best for all models. Has any analysis been done to ensure these aren't overfitting? This could be interesting to explore with Table 14 and/or figure 8, but table 14 without this doesn't seem overly informative.

**Response 27**: Thank you for your comment. We do not have overfitting and we have improved its explanation. As shown in Section 4, in particular Figure 7 in the revised version, we do not have overfitting because the error difference between using 90% and 60% of the data for training (the maximum and minimum percentages, respectively) is approximately 2% in the worst-case scenario (1-hour dataset). This suggests that overfitting is not significant in the proposed model. Thus, this clarification is given in lines 281-284, as follows:

behaviors are shown by the other models, in particular with ADA model. **Regarding overfitting, Figure 7 shows that the error difference between using 90% and 60% of the data for training (the maximum and minimum percentages, respectively) is approximately 2% in the worst-case scenario (1-hour dataset). This suggests that overfitting is not significant in the proposed model.**

In addition, notice that in Section 4, the metrics presented in Tables 9-11 for the different models and training-test ratio datasets represent the weighted average over 100 iterations. In each iteration, the content of the training and test sets is varied to obtain results with minimal bias. This detail is also included, as shown below (lines 255-262):

255 **4 Results**

We evaluated the performance metrics of these ML models under different configurations (in terms of $R^2$, RMSE, MAE and Mean Absolute Prediction Error (MAPE) in $\mu g/m^3$ and execution time in seconds), both with default and optimized hyperparameters, taking into account the three different datasets given by different monitoring intervals: 10 and 30 min and 1 h, as depicted in Section 3.2. We tested different training-test ratio percentages from these datasets: 60%-40%, 70%-30%, 260 80%-20% and 90%-10%, denoted as 60/40, 70/30, 80/20 and 90/10. **The metrics presented in the Tables 9, 10 and 11 are the weighted average of each metric over 100 different iterations by changing the content of the training and test set to obtain results with the minimum bias as possible.**

Similar with table 15 – in line 224, the authors write, "In the same line as before, once again we can see how the GB adjusts better compared with the other models." If this entire

table exists just to make a point about GB that has already been made, is it a necessary table?

**Response 28**: In the new version, we emphasize that the values in Table 12 are derived from the different error distributions shown in Figure 8. This information offers a different and complementary perspective (rather than calculating R² and the various errors), as it considers the Standard Deviation (SD, σ) and the Confidence Interval (CI). With this approach, we observe that the Gradient Boosting (GB) model performs better compared to the other models based on the error distribution analysis. We have also improved the wording related to Table 12 and Figure 8 in lines 293-294 to better link both results, highlighting that the SD is similar to the RMSE. This is due to the fact that, as shown in Figure 8, the distribution is nearly Gaussian. This is the improved paragraph (lines 293-294):

> The Standard Deviations (SD) and the Confidence Intervals (CI) in $\mu g/m^3$ are shown in Table 12. This information is obtained from the error distribution statistics given in Figure 8. It can be seen how the GB adjusts better compared with the other models from the error distribution analysis. **It is observed that the SD is similar to the RMSE and this is due to the fact that, as shown in this figure, the distribution is almost Gaussian.**

In tables 6, 7, 12, 13, 14, 16, 17, I suggest clarifying in the captions whether this is training or testing data. If it's all training data, I would be very interested in seeing the training data added as the training data is a better indicator of how this model would actually perform in the field.

**Response 29**: Note that all the data in these tables are test data, not training data. The training data from various iterations are used to create models, which are then evaluated using new data, referred to as test data. Metrics are extracted from the results obtained using this test data.

Figure 7 is great and the most informative in the paper. If a graphical abstract is requested, I would suggest this one.

**Response 30**: Thank you.

Figure 8 has a typo in 'Percentage' on the lower right. This plot could be much stronger if the R2 and RMSE were plotted for both the training and testing data instead of just training. It's no question that the more training data you have, the better the fit will be – the training data is what will indicate whether you're overfitting. This would be a great place to address overfitting in your discussion.

**Response 31**: We have fixed this type. As we explained in Response 29, all this information is from testing, not training. About overfitting, this was already answered in Response 27 based on the information from Figure 7. Also, it is clarified in the new version of the manuscript in lines 281-284.

Figure 9 could use a sentence in the accompanying paragraph (starting at line 217) plainly stating what they key takeaway should be. Is it that HOP improves the model greatly regardless of the original model used?

**Response 32**: Thank you for this comment. Yes, HOP optimizes the different models. As is explained in the manuscript in Section 3.4 (lines 211-218), the goal of HPO on these algorithms is to adjust the hyperparameters as a tuning technique that exhaustively searches through a user-defined hyperparameter space to find the optimal combination. These hyperparameters are external specific model configurations settings, aiming to identify the configuration that maximizes estimation accuracy given by $R^2$, RMSE and MAE.

Then as a result, when HPO has been applied on the different models, the distribution error (as shown in Figure 8 in the new version) tends to concentrate around 0, in a remarkable way for the GB and ADA but with practically no change for the DT and RF.

This explanation has been introduced in Section 4 (Results), next to this Figure 8, in the revised version as follows:

285    In Figure 8, it is shown the distribution error for the different models, with detail of raw, default and optimized versions. The number of samples are normalized in the Y-axis. It is appreciated with GB and ADA that their distribution errors are concentrated around zero when calibration is applied, and even more when using the HPO optimized models. This behavior is also appreciated with DT, but with lower intensity. However, RF keeps a pretty similar distribution in both versions, default and optimized, as we saw in Tables 9 and 10. **Thus, in terms of error distribution, HPO significantly concentrates the error** 290    **around 0 for the GB and ADA models, while practically no change is observed for the DT and RF models.**

I'm not sure I see the value of table 16 – as you stated in the introduction and in Figures 4-5, the raw sensor readings are completely unreliable on their own.

**Response 33**: The goal of this table (Table 13 in the revised version) is to quantify the improvement vs the raw readings from the low-cost ozone sensor of the ZPHS01B module. In other words, it shows the gain in terms of $R^2$ and error reduction when applying the ML techniques over the raw readings.

In table 17, 'et al (2016)' seems to be missing a name. I'm not sure I see the value of table 17 – if these were other studies comparing ozone quantification with the ZPHS01B module, that would make more sense to me than seemingly randomly selected projects using different sensors at different price points?

**Response 34**: Thank you. We have fixed this reference in Table 14 (in the revised version). About the content of this comparison table, we compare our models for O3 calibration against the related work with a similar approach. As we state in the manuscript, we stress that the starting point of these studies are slightly different, since they have used more reliable and expensive low-cost sensors, approximately ten times more expensive than the ZPHS01B module. Thus, the comparison is for the whole system and not only the algorithm. It is worth mentioning that there are no other studies using this module. Our model reduces the estimation error up to 94.05% from raw readings with a Mean Relative Error (MRE) of 7.21%, given by MAE 4.022 with 90/10 dataset and with O3 mean value of 55.72 µg/m³ using Gradient Boosting (GB) with only 4 features.

We have improved the wording of this paragraph as follows (lines 297-302):

In Table 14, we compare our models for O3 calibration for LCS, against the related work with a similar approach. We must stress that the starting point is slightly different compared to ours, since these studies have used more reliable and expensive LCS, approximately ten times more expensive that the ZPHS01B module. As mentioned before, our model reduces the estimation error up to 94.05% from raw readings based on MAE measurements, with a MRE of 7.21% (given by MAE 4.022 with 90/10 dataset and with O3 mean value of 55.72 $\mu g/m^3$ as shown in Table 3, using GB with only 4 features, as shown in Section 3.3.

This study builds a strong ML model to fit a single low-cost sensor for ozone. Some of the challenges regarding field deployments of low-cost sensors include ensuring that each individual node is properly calibrated, and that these calibrations perform just as well in the field, where temperatures, humidities, and ozone concentrations not seen during the co-location with a reference instrument appear. For future works, it would be great to see the authors address what their path forward might look like.

**Response 35**: Thank you for your comments. We have revised the wording regarding the related work as follows (lines 314-317):

As future work, we plan to expand the dataset and include complementary parameters, such as wind speed or road traffic density, to increase the accuracy of these models. In addition, we highlight that our research activity is focused on the design of new calibration and forecasting algorithms for the different sensors embedded in the low-cost ZPHS01B module in order to improve AQ monitoring resolution.

Thank you for your thoughtful review and comments which will enable us to improve this work. We appreciate the time and effort invested in your review.

---

## Author Comment (AC2)

**REFERENCE: amt-2024-127- "CC1"**

**Title: "**_Improving Raw Readings from Low-Cost Ozone Sensors Using Artificial Intelligence for Air Quality Monitoring_"

**Authors:** _Guillem Montalban-Faet, Eric Meneses-Albala, Santiago Felici-Castell, Juan J. Perez-Solano and Jaume Segura-Garcia_

Departament de Informàtica, ETSE, Universitat de València, Avd. de la Universidad S/N, 46100 Burjassot, (Valencia), Spain

Dear editor and reviewer,

Thank you for giving us the opportunity to address the comments provided by the anonymous reviewers. We have made every effort to respond thoroughly to their feedback. Attached is a response letter with our responses highlighted **in blue**. The revised manuscript also uses **blue text** to indicate the changes made. In some answers, this **blue text** is highlighted if there is more than one answer.

We would also like to express our gratitude to the anonymous reviewers for their valuable comments and suggestions. We appreciate the time and effort they have invested in improving our work. We firmly believe that this manuscript is now suitable for publication and an excellent contribution to share with the broader research community.

**Reviewer's comments (A. Kourtiche Referee #2 CC1, 24 Jan 2025)**

The article focuses on leveraging low-cost sensors (LCS), specifically the ZPHS01B module, to monitor ground-level ozone ($O_3$), a critical air pollutant and urban pollution indicator. Due to the limited accuracy of LCS, the authors applied advanced Machine Learning (ML) methods, including Gradient (GB), (RF), (ADA), and (DT), for calibration.

The dataset spans 165 days, with optimal results obtained using a 10 min monitoring interval. The GB model achieved the best performance, reducing the estimation error by 94.05%, while other models reduced errors by more than 89%. (HPO) and feature selection techniques (FIA, FS) improved model performance.

First of all, we would like to sincerely thank you for your thoughtful review and comments, which have greatly contributed to improving our work.

In the following sections, we will address all your comments, queries, and suggestions. This is an extended version of an answer provided to, we guess the same reviewer, as RC2 from Feb 4 2025 about the same manuscript.

The authors plan to extend the dataset and include additional parameters like wind speed and traffic density in future work.

I.Questions:

1. Why was a 10 min interval more effective than 30 min or 1 hour ? Could other time intervals (e.g., 5 or 15 minutes) be explored?

**Response 1:** Thank you for this comment. 10 minutes was the minimum interval given by the references, since it is a standardized monitoring interval for outdoor official air quality monitoring in normal conditions as depicted in Section 3.2.

And among the datasets given by the 10 min, 30 min and 1-hour intervals, after training the models as it is explained in Section 4, we see that the minimum prediction error is achieved by the 10 minutes interval, as it is shown in Figure 7 for RMSE and MAE in the new version of the manuscript (Figure 8 in the first version).

Notice that this is due to several reasons. On one hand, the 10 min interval is determined by environmental researchers as the interval that better gathers the different outdoor air quality behaviors, with higher detail under normal conditions. Higher sampling frequencies (lower monitoring intervals) create oversampling and redundancy. On the other hand, if we use higher monitoring intervals (30 min or 1 hour), we see that

we start losing details, smoothing the dataset and overlooking different behaviors that in the Machine Learning (ML) process helps to reduce the prediction error.

We have clarified this issue in the new version of the manuscript as follows:

> In particular, we have used a data set of 165 days, with different monitoring intervals, giving the best results when we use 10 min monitoring interval, as it could be expected. **If we use higher monitoring intervals (30 min or 1 hour), we see that we start losing details, smoothing the dataset and overlooking different behaviors that in the ML process helps to reduce the prediction error.** For the training process, we have carried out several techniques (FIA and FS) in order to select the most

330

2. Were the ensemble models compared statistically to determine significance in performance differences?

How do these models generalize to new datasets or different geographical locations?

**Response 2:** Yes, these models were tested and compared in Section 4. We evaluated the performance metrics in terms of R^2, RMSE, MAE and MAPE as shown in Tables 9-11. These results are the weighted average of each metric over 100 different iterations by changing the content of the training and test set to obtain results with the minimum bias as possible as explained in the manuscript. This explanation is included in the new version as follows:

> Notice that the performance metrics shown in Tables 9, 10 and 11 are the weighted average of each metric over 100 different iterations by changing the content of the training and test set to obtain results with the minimum bias as possible.

270

With regard to the generalization of different datasets, this is considered by taking a sufficient dataset, as it is detailed in the reference "Machine Learning in Environmental Research: Common Pitfalls and Best Practices" by Zhu, et al. In particular, as it is explained in Section 3.2, the recommended relationship (ratio) between Sample size and Feature size (Sample-size to Feature-size Ratio (SFR)) is higher than 500. In our datasets, we have a sample size of 23496, 7843 and 3922 for 10 min, 30 min and 1 hour interval, that is a SFR of 4699.2, 1568.6 and 784.4, since we only use 4 features, as it is depicted in Section 4.

About extending the dataset with more data, notice that the fusion of the different datasets from different locations as a first approach is not recommended, since they could change the environmental conditions. This merging process would require refinement in the datasets as well as in the models, that in this case, given the available datasets are not necessary. It is better to work with different datasets from different locations separately, independently.

Nevertheless, in order to answer the reviewer, we have created another dataset (Dataset 2) with new samples from another deployment with two different LCS nodes (called AQ IoT Node 1 and 2) in a different location, in Valencia city. In particular, the new dataset is from the official AQ monitoring station called Moli del Sol (Valencia, Spain) placed at 39.48113875, -0.40855865, managed by Generalitat Valenciana (GVA)

and its data is retrieved from https://rvvcca.gva.es/estatico/46250048, for O3 calibration. This station is 4.1 km away from the previous official station used for the dataset in the manuscript. In this case, this dataset is from May 31, 2024, till January 25, 2025, it has 239 days and includes data from different seasons as suggested by the reviewer. Notice that in our case, to carry out all these deployments, it is required to ask for permission to the official institutions in charge of Air Quality.

Thus, with this new dataset (Dataset2), we have repeated the same process as explained in the manuscript, achieving nearly the same results as shown below. We show the HPO results over 100 different iterations by changing the content of the training and testing set (with the best results given by 90%/10% ratio as already discussed in Section 4) to obtain results with the minimum bias as possible, for both nodes (AQ IoT Node 1 and 2):

**NODE 1**

GradientBoostingRegressor(criterion='squared_error', max_depth=None, learning_rate=0.1,max_features=1.0, n_estimators=900, subsample=1.0)
R2 = 0.9405841973910234
RMSE = 6.107097433579371
MAE = 4.336455961006405
MAPE = 0.1679585719053396
time = 102.18236994743347

RandomForestRegressor(max_depth=None,max_features=1.0, n_estimators=100)
R2 = 0.9046692614127114
RMSE = 7.735712909738179
MAE = 5.23282966066717
MAPE = 0.20992345469839893
time = 27.86010217666626

AdaBoostRegressor(estimator=DecisionTreeRegressor(max_features=1.0),
 n_estimators=50, learning_rate=0.01, loss='exponential')
R2 = 0.9090424941272316
RMSE = 7.556194639834324
MAE = 4.564039465946062
MAPE = 0.16874010491994965
time = 11.807359457015991

DecisionTreeRegressor(max_depth=None, max_features=1.0, splitter='best')
R2 = 0.8191113924173187
RMSE = 10.655883718994565
MAE = 6.295906305813436
MAPE = 0.2235395149139127
time = 0.33399152755737305

**NODE 2**

GradientBoostingRegressor(criterion='squared_error', max_depth=None, learning_rate=0.1,max_features=1.0, n_estimators=900, subsample=1.0)
R2 = 0.9547003457380135
RMSE = 5.332505456267162

```
MAE = 3.7416539078656776
MAPE = 0.14152286529664848
time = 82.03594541549683

RandomForestRegressor(max_depth=None,max_features=1.0, n_estimators=100)
R2 = 0.934358633720318
RMSE = 6.419078264005403
MAE = 4.187047365360581
MAPE = 0.15794878544527777
time = 20.572300910949707

AdaBoostRegressor(estimator=DecisionTreeRegressor(max_features=1.0),
 n_estimators=50, learning_rate=0.01, loss='exponential')
R2 = 0.9287003904552755
RMSE = 6.690020376986309
MAE = 3.8299511364469465
MAPE = 0.13586980540686971
time = 8.766397953033447

DecisionTreeRegressor(max_depth=None, max_features=1.0, splitter='best')
R2 = 0.8745869394552789
RMSE = 8.872688771740032
MAE = 4.974713868475632
MAPE = 0.16654468197115013
time = 0.23625636100769043
```

As seen in this new dataset, both AQ IoT nodes exhibit similar behavior. However, Node 2 performs slightly better than Node 1, likely due to manufacturing variations associated with their low cost. It is important to emphasize that these results closely resemble those already presented in the manuscript. In the following table we compare and summarize these results from *Dataset 1*, the one used in the manuscript, and *Dataset 2*, the new data set analyzed here in the review.

| GB optimized | Dataset1 | Dataset2 (Node1) | Dataset2 (Node2) |
|---|---|---|---|
| $R^2$ | 0.938 | 0.940 | 0.954 |
| RMSE | 6.492 | 6.107 | 5.332 |
| MAE | 4.022 | 4.336 | 3.741 |
| MAPE | 0.194 | 0.167 | 0.141 |
| Time [s] | 66.937 | 102.182 | 82.035 |

 As we can see, Node 1 works worse than Node 2, and the previous results obtained from Dataset1 are between these two. In this case, with Dataset 2, the Mean Relative Error (MRE) is 6,71% for Node 2 and for Node 1 is 7.78%, and with Dataset 1 was 7.21%. The estimation of the MRE discussion is at the end of Section 4 in the new version of the manuscript as follows:

the estimation error up to 94.05% from raw readings based on MAE measurements, **with a MRE of 7.21% (given by MAE 4.022 with 90/10 dataset and with O3 mean value of 55.72 $\mu g/m^3$ as shown in Table 3**, using GB with only 4 features, as shown in Section 3.3.

Thus, based on this information, we conclude that for the ZPHS01B module, 165 days of Dataset 1 provide sufficient information to generalize the proposed calibration models. This aligns with the SFR recommended values, as stated earlier. In other words, given the features and characteristics of this module, the original dataset (165 days) contains enough information to generalize the behavior of the O3 sensors and their response. Thus, better results cannot be achieved with other datasets given the constraints of this module.

This information has been included in the new version of the manuscript with these modifications, in Section 3.1, describing the dataset-2 as follows:

> In addition, in order to test the proposed models in this paper and their generalization in Section 4, we have used another dataset (dataset-2) with two different AQ IoT nodes (Node 1 and 2), from the official AQ monitoring station
> 160  called *Moli del Sol* (Valencia, Spain) with latitude and longitude 39.48113875, -0.40855865. This station is 4.1 km away from the previous one. Its data is retrieved from *https://rvvcca.gva.es/estatico/46250048*. In this case, this dataset is from May 31, 2024 till January 25, 2025, with 239 days. Now on, we will refer always to dataset-1 as the dataset, except in Section 4 where we generalize the models with dataset-2.

In Section 4, in the results with:

**Table 13.** Generalization test with dataset-1 and dataset-2 (Node 1 and 2) using GB$_{optimized}$ algorithm with 90/10 (training/testing) ratio.

|  | Dataset-1 | Dataset-2 (Node 1) | Dataset-2 (Node 2) |
|---|---|---|---|
| $R^2$ | 0.938 | 0.940 | 0.954 |
| RMSE | 6.492 | 6.107 | 5.332 |
| MAE | 4.022 | 4.336 | 3.741 |
| MAPE | 0.194 | 0.167 | 0.141 |

> In terms of generalization as mentioned in Section 3.1, we have checked the same proposed models with dataset-2 under the same conditions, with 90/10 (training/testing) ratio. In Table 13, we summarize the metrics given by the best model based on GB for dataset-1 and for Node 1 and 2 from dataset-2 respectively. In particular, if we focus on MAE,
> 310  we see that Node 2 performs slightly better than Node 1 in dataset-2, likely due to manufacturing variations associated with their low cost, as well as the results from dataset-1 are between these two, validating its generalized behavior.

As well as in the conclusion section:

> Besides, we checked that for the ZPHS01B module and O3 calibration, 165 days of dataset-1 provided sufficient information to generalize the proposed models comparing with a dataset-2 of 239 days. This aligns with the SFR recommended values according to (Zhu et al. (2023). Thus, given the features and characteristics of this module, the original
> 335  dataset (165 days) contains enough information to generalize the behavior of the O3 sensor and their response.

3. Why were only 4 features used for the GB model? Were additional environmental factors considered initially but excluded?

**Response 3:** Thank you for your comment. Based on the analysis conducted in Section 3.3 for feature selection, we proceeded in Section 4 with the features that provided the best performance metrics. These selected features are [date, O3, T, RH], as initially indicated in Section 4. We omitted other features that led to poorer results. It is important to note that adding less significant features can reduce the importance of key parameters, ultimately affecting the overall performance. For instance, including both T and PM results in worse performance compared to using only T, leading to less effective models.

4. How does the proposed approach balance cost savings with performance?

**Response 4:** Low-cost sensors, as detailed in Section 2, are much cheaper than official equipment but with lower accuracy.

Taking this information into account, Table 15 of the manuscript (in the new version and 17 in the first one), compares our approach to O3 calibration with similar related work. It is important to note that the starting point of the selected studies for comparison differs slightly from ours, as these studies used more reliable and significantly more expensive low-cost sensors—approximately ten times the cost of the ZPHS01B module as depicted in Table 1 of Section 2 with its price range. This was already included in the manuscript.

| Module | Sensors | Price range |
|---|---|---|
| SDSO11 (Nova Fitness Co., Ltd. (2024)) | Temp, RH, PM, PA | Low |
| DL-LP8P (DecentLab, Ltd. (2024)) | Temp, RH, CO2, PA | Low |
| MiCS-6814 (SGX, SensorTech (2024)) | CO, NO2, C2H5OH, NH3, CH4 | Low |
| ZPHS01B (Zhengzhou Winsen Electronics Technology Co. (2024)) | Temp, RH, PM1-10, CO, CO2, O3, NO2, TVOC | Mid-Low |
| Sensit RAMP (Sensit (2024)) | PM2.5, CO, CO2, NO, NO2, O3 | High |
| AirSensEUR (Van Poppel et al. (2023)) | NO, NO2, O3, CO, PM2.5, PM10, PM1, CO2 | Mid-High |

**Table 1.** AQ Sensor modules with cost estimate: Low (less than 10$), Mid-Low (100-200$), Mid-High (600-1000$) and High ($\approx$<4000$).

Thus, we consider that this is a fair balance, highlighting the improvements for the O3 calibration by using this module.

II.Improvements Needed:

The current dataset covers 165 days. Increasing the dataset size and covering different seasons or regions could improve generalizability.

**Response 5:** Thank you for this comment.

As already answered in Response 2, we repeated the analysis with a new dataset (Dataset 2) from May 31 2024 till January 25, 2025. This dataset has 239 days and includes data from different seasons as suggested by the reviewer.

As a first approach, we agree that increasing the dataset size and covering different seasons or regions could improve the generalizability of ensemble machine learning algorithms. In general, a larger dataset typically provides more diverse examples, allowing the model to learn from a wider variety of patterns and reducing the risk of overfitting to specific data characteristics. Even when the data spans different locations, the model also may become more robust by capturing the seasonal variations and regional trends that might not be present in a smaller, localized dataset.

However, as stated in Response 2, 165 days of Dataset 1 provided sufficient information to generalize the proposed O3 calibration model and better results cannot be obtained with other datasets given the constraints of this module.

Besides, it must be stressed that there is a trade-off between accuracy and life-time of the low cost sensor. And this is the main reason we cannot last the different deployments for years. In particular, these low cost sensor modules degrade fast and their accuracy is reduced in months.

Adding complementary parameters, such as traffic patterns, industrial activities, and meteorological conditions, could enhance the model's robustness.

**Response 6:** Thank you for your interesting comment. Although this approach is very interesting and valid for some scenarios, in our case we focus only on Air quality information obtained directly from the low-cost sensor modules. Of course we could include other related information in more theoretical studies, but not in a real scenario as the one proposed. This type of information (traffic patterns, industrial activities) is not available easily in real time, assuming the low cost IoT AQ nodes, described in this paper. As it is explained in Section 2, usually, these nodes have limited communications and only can gather local information from their directly connected sensors. And when the information is processed, they can run the ML models to improve the accuracy of the readings. Finally, they can upload this information to other external servers, but always with constraints due to their features.

Besides, other meteorological sensors (such as wind speed and direction) could be interesting, but in the end they will modify the different diffusion models of the different gasses, but in practice they do not alter the direct readings of the low cost Air quality sensors, if they are properly housed as we did in deployment.

Nevertheless, this discussion has been included in Section 5 in the conclusion as future work, but more focus on theoretical studies rather than on real deployments with constrained devices as the ones used for Air quality monitoring with low-cost features.

-While GB is identified as the best-performing model, a statistical comparison of model performances (e.g., paired t-tests on errors) should be included to support conclusions.

+Explain why ADA and RF performed similarly or differently from GB.

-Discuss the trade-off between GB's higher execution time and its improved accuracy.

-Propose optimizations for deployment scenarios requiring real-time predictions.

**Response** 7:  Thank you for your comments. Next, we provide an extended explanation about these issues. The different key points about this explanation have been used in order to improve the wording in different parts of the manuscript.

Find next a detailed discussion about all these items.

As we mentioned below, the guidelines to process this kind of data is shown in reference "Machine Learning in Environmental Research: Common Pitfalls and Best Practices" by Zhu, et al.. Thus, in particular, about the mentioned "paired t-tests on errors", these tests are used to test if the means of two paired measurements are significantly different, but this does not apply in our experiments, since the different models are carried out independently and using different data-sets, as it is explained in Section 4 and different "training-test" ratio percentages from these datasets: 60%-40%, 70%-30%, 80%-20% and 90%-10%. Besides, during the training process, each performance metric depicted in Section 4 based on $R^2$, RMSE, MAE and MAPE is obtained with 100 different iterations by changing the content of the training and test set to obtain results with the minimum bias as possible.

About the behavior of ADA and RF vs GB, although all of them are ensemble ML algorithms, their algorithms are based on slightly different approaches. In particular, as explained in Section 3.4, Adaptive Boosting (AdaBoost) and Gradient Boosting differ in how they improve performance. AdaBoost focuses on re-weighting the training data, assigning higher weights to misclassified examples, so subsequent weak learners focus on these harder cases. It combines weak learners using weighted voting, emphasizing the most accurate ones. In contrast, Gradient Boosting focuses on minimizing a specific loss function by fitting each new weak learner to the residual errors (differences between actual and predicted values) of the previous model. This makes Gradient Boosting more flexible, allowing it to handle custom loss functions and more complex learners. While AdaBoost is simpler and faster, but sensitive to noise, Gradient Boosting is more powerful and robust for complex tasks, but it requires higher execution time.

Similarly, Random Forest and Gradient Boosting are both ensemble learning algorithms and use decision trees as base models, but differ significantly in their approach. Random Forest builds multiple decision trees independently by randomly sampling data and features, then aggregates their predictions (via majority vote for classification or averaging for regression). This parallel training makes it robust, fast, and less prone to overfitting. In contrast, Gradient Boosting trains decision trees sequentially, where each tree attempts to correct the residual errors of the previous ones by optimizing a specified loss function. This iterative process makes Gradient Boosting more flexible and capable of fine-tuning but slower. While Random Forest excels in robustness and simplicity, Gradient Boosting often achieves higher accuracy in complex tasks due to its ability to learn from mistakes adaptively.

Besides, these algorithms (ADA, RF and GB) have different hyperparameters and with different optimized values, adjusted independently by HPO techniques, as shown in Table 5-8 in Section 3.4.

Regarding the execution time, the trade-off between Gradient Boosting's higher execution time and its improved accuracy compared to Adaptive Boosting and Random Forest comes down to the balance between computational cost and predictive performance. Gradient Boosting builds trees sequentially, optimizing a specific loss function at each step, which allows it to capture complex patterns and often achieve superior accuracy. However, this iterative process makes it computationally intensive and slower, especially for large datasets or when fine-tuning hyperparameters. Adaptive Boosting, while also sequential, is generally faster because it uses simpler learners (like decision stumps) and focuses on re-weighting misclassified points rather than optimizing a loss function as mentioned before. Random Forest, in contrast, trains trees independently and in parallel, making it much faster, but it sacrifices some accuracy because it relies on averaging predictions instead of iterative error correction. While Gradient Boosting excels in tasks where accuracy is paramount, its higher execution time may not be justified for less complex problems or time-sensitive applications, where Random Forest or Adaptive Boosting could provide a faster, more practical solution.

And finally, about the optimizations to be applied on the deployments for real-time predictions, it must be stressed that once these models are trained, they can be ported to the low cost AQ node that is based on a microcontroller. Then, with these models we can improve the accuracy of the direct readings immediately.

Notice that these details have been used to enrich the new wording in Section 4 when dealing with the different algorithms. Besides, it has been included in the future work, since in practice, this is a very interesting point for the whole AQ monitoring network.

III-Proposed Best Method

-Explore DL models like LSTMs or Temporal Convolutional Networks (TCNs) for time-series prediction to capture long-term dependencies.

**Response 8:** Thank you for this interesting comment.

Gradient Boosting algorithms are often more practical, efficient, and interpretable for time-series prediction tasks, especially when datasets are small-to-medium-sized, contain noise, or require explicit domain knowledge. While DL models like LSTMs and TCNs excel in capturing long-term dependencies in very large datasets, Gradient Boosting flexibility, lower data requirements and ease of use make it a strong choice for real-world time-series applications.

Nevertheless, as it is mentioned before, the lifetime of these low cost sensors and their performance degrade over the time (aging), due to their manufacturing process. In particular, this is more critical in the ZPHS01B module and that is the reason we focused on these ensemble algorithms.

Of course, there is a tradeoff between ML and DL in these scenarios, but pros and cons made us conduct the test with these ML techniques, with good results.

It is worth mentioning that we also used DL techniques, but we observed that they are not able to generalize as the ML approach did. And for this reason, the results using DL techniques are not so robust and reliable, mainly due to overfitting even with bigger datasets in this context and scenario. These results are shown below for a simple Sequential Neural Network from TensorFlow/Keras using an optimizator *stochastic gradient descent* with an input of 4 features and two layers. These two layers are a dense layer with four neurons and a linear activation, followed by a second layer with a neuron that provides the output. The network scheme is shown in **Figure 1**, below.

[Figure]

*Figure 1:* *Scheme of the Sequential Neural Network from TensorFlow/Keras using an optimizator stochastic gradient descent with an input of 4 features and two dense layers.*

The results from this *Sequential Neural Network* are:

**R2 Score: 0.9999999999976741**

**RMSE: 3.514481801502949e-05**

**MAE: 2.9925663790820442e-05**

As we can see, these results show that these techniques learn and memorize the whole dataset and we cannot generalize. That is the reason we focused on ML since they adapt and perform better in this scenario, given by the AQ monitoring stations and the ZPHS01B low cost sensor module for O3 calibration.

We have included this explanation also in the revised version, in order to justify the selection of these ML techniques instead of other techniques. This information is included as follows:

**210  3.4  Applying Machine Learning algorithms**

As mentioned before in environmental research, the use of ML algorithms, in particular ensemble models, has increased significantly compared to DL (Zimmerman et al. (2018)). Some of the most popular ensemble algorithms are RF or GB related models (Obregon and Jung (2022)). **Furthermore, based on our experience, we recognize that in AQ monitoring scenarios using LCS such as the ZPHS01B module, datasets are often limited and constrained, which affects the use of**

215  **DL techniques, as they usually tend to overfit.**

-Combine GB with DL methods for feature extraction and refinement, especially if additional parameters are included.

**Response 9:** Thank you for this comment.

From our experience, combining Gradient Boosting algorithms with Deep Learning methods for feature selection is often unnecessary due to several reasons. Gradient Boosting algorithms as the ones proposed in this research, are inherently capable of handling feature selection through their built-in mechanisms, such as calculating feature importance and automatically ignoring irrelevant or redundant features during training as it was shown in Section 3.4. These algorithms excel in structured data tasks and effectively model complex, non-linear relationships without requiring additional feature selection methods as depicted in Section 3.4. Furthermore, Deep Learning-based feature selection is computationally expensive, requiring significant resources and larger datasets to avoid overfitting, which may not justify the effort when Gradient Boosting can already achieve competitive results. Additionally, Gradient Boosting provides interpretable outputs which offer clear insights into feature importance, unlike Deep Learning methods, which often function as black boxes. Finally, introducing Deep Learning adds unnecessary complexity to the pipeline, increasing training time and resource demands without guaranteed improvements in predictive performance, especially when Gradient Boosting already performs well on the given dataset.

This explanation and justification have been considered in the new version of the manuscript in Section 3.4.

-Use advanced ensemble techniques like Stacked Generalization (Stacking) to blend predictions from GB, RF, and ADA for better accuracy.

**Response 10:** Thank you for this comment.

It must be pointed out that using Stacked Generalization (Stacking) to blend predictions from Gradient Boosting, Random Forest and AdaBoost may not be ideal due to several reasons. First, it adds complexity by introducing a meta-learner, making the workflow harder to interpret and manage, often for marginal accuracy gains. Gradient Boosting already iteratively optimizes predictions and often outperforms combinations with simpler models like Random Forest or AdaBoost, making the stack redundant. Additionally, stacking increases the risk of overfitting, especially with small datasets, as the meta-learner can overfit to the base models' predictions. It also significantly increases training time and computational demands, while the lack of diversity among tree-based models reduces the potential benefits of combining them. Simpler alternatives, such as weighted

averaging or selecting the best-performing model, often achieve comparable results without the added complexity.

Several scientific references support the arguments against using Stacked Generalization (Stacking) to combine predictions from Gradient Boosting, Random Forest, and AdaBoost. The increased complexity and risk of overfitting associated with stacking are highlighted in "A guide to ensemble learning," which notes that ensemble methods can lead to computational complexity and overfitting risks [1]. Additionally, the article "Stacking to Improve Model Performance: A Comprehensive Guide" discusses how utilizing too many base models in a stacked ensemble can result in overfitting and increased computing complexity [2]. Furthermore, the article "Gradient Boosting vs Random Forest" explains that Gradient Boosting focuses on sequential correction of errors, while Random Forest relies on the diversity of independently trained trees, suggesting that combining these models may not provide significant additional benefits [3].

Thus, based on these reasons, it shows that for this case, stacking these particular models may introduce unnecessary complexity and overfitting risks without substantial improvements in predictive performance.

However, this comment has been included in the new version of the manuscript, to justify this explanation.

*References*:

[1] https://serokell.io/blog/ensemble-learning-guide

[2] https://medium.com/@brijesh_soni/understanding-boosting-in-machine-learning-a-comprehensive-guide-bdeaa1167a6

[3] https://www.geeksforgeeks.org/gradient-boosting-vs-random-forest/

IV-Recommendations :

Expand the dataset and include more parameters to increase model accuracy.

Conduct real-world validation to demonstrate scalability and robustness.

Compare ML and DL approaches to assess their suitability for time-series AQ calibration.

Provide open-source tools for replicating and extending the proposed calibration process.

By addressing these improvements and exploring advanced methodologies, the study can significantly contribute to cost-effective and scalable air quality monitoring solutions.

**Response 11:** Thank you for feedback.

We consider that all these issues have been discussed during this review, and some of them, the more interesting, have been included in the new version of the manuscript improving its wording.

In summary, about the dataset, we have discussed this in Response 2 and 5 with detail, as well as using other locations. About the real-world validation, all our trials and measurements come from real deployments. We have not used anything simulated. About the comparison between ML and DL, as it was discussed previously, we have included this discussion as well as their worse results, in favor of ML in this case. Also, about the open-source tools, all our datasets are available online, as it is indicated in the last part of the manuscript with the following statement "Please feel free to contact to the authors for further information: http://www.uv.es/eco4rupa/dataset.html".

Finally, thank you for your thoughtful review and comments which will enable us to improve this work. We appreciate the time and effort invested in your review.

---

## Referee Report (RR1)

Summary: While this draft shows improvement, more work is needed on the introduction/related work to set the stage for a strong paper. These sections should clearly set up: 1) what is already being done in the space; 2) what is lacking in the space; 3) what you will do differently to expand on what's already been done. There are plenty of other papers already using ML, GB., etc. – what about your model is different? Likewise, figures and tables should be included selectively – many are still superfluous and either demonstrate the same information as each other, or information that is already well-established in the field. These figures should be combined or removed as appropriate.

Subscripts are needed throughout for $O_3$, $CO_2$, etc.

Abstract: Readers will know what ozone is. This space would be better spent explaining why you need machine learning enabled calibration.

Line 14: do the authors ever come back to these guidelines? If not, this paragraph is not useful. Same with the next paragraph – these standards are not really mentioned again later. I understand that this is trying to establish the "why use low-cost sensors", but it needs to be more clearly related back to what you're actually doing.

Line 27: Why is "primary" in quotes but not secondary? Be consistent, but quotes are not necessary. There are also quotes around primary on line 2.

Line 34: What is "official equipment"?

Line 51: While not detrimental, this paragraph is unnecessary.

Related works: see comments about table 15, but the information on the specific other sensors used for comparison could be restructured, if not removed. The information added here on specific ML models here is helpful, but it could be improved further by exploring more clearly the strengths and weaknesses of each of these, and how you will improve upon this and not just repeat what's already been done.

Table 3: This table should be removed. There are still no units in this table (temp, RH, PM2.5, CO2, NO2, CO, etc. should all have units attached). The statistics of the measured quantities are not referenced or used anywhere else in the paper, and the reader can't do anything with this information on their own. Likewise, "stationarity" and "percentage of samples taking Different values" are not analyzed further in the text. The paragraph beginning on line 181 can be condensed to give the context the table is hoping to provide (ex. "Sensors X, Y, and Z appeared particularly unreliable and were omitted from our model".)

Figure 3: Any ambient pollutant will have a repeating diurnal pattern from the boundary layer rising and falling each day, and most sensors will pick up on major sources like traffic.

A DFT is not necessarily needed to show this and confuses the messaging in this section. Since this figure is never referenced again other than to show that a pattern exists, it should be omitted.

Figure 4: While this figure is fine, it's well known in the low-cost sensor space that sensors can capture the general trends of pollutants but need calibration to accurately convey the magnitude. This figure should be omitted.

Table 4: If this is all to make a better ozone model, the FIA of ozone should be included here to show how much it improves the model. How was 8% importance selected? It sounds arbitrary. It would also be easier on the reader if the threshold and the table were in the same format (either both in decimal or both in percent).

Figure 5 is essentially showing that some sensors are more cross sensitive than others, which is already well established in the field. This figure should be omitted.

Tables 5, 6, 7 and 8 should be combined into a single table with the 4 sub-categories as another column.

Table 9 is unnecessary and can be omitted – you and many others have already established that hyperparameter tuning will make the models fit better.

Tables 10 & 11 should also be combined.

For tables 9, 10, and 11, and Figure 6, it is not specified in the titles whether it is training or testing data – please specify.

In the low-cost sensor field, it is standard to show both training and testing data - consider adding to tables 9, 10, and 11, and Figure 6.

Is the point of Figure 7 just to show that the model isn't overfitting? It needs more analysis in the text rather than relying on the reader to interpret.

Again, it is well-established that low-cost sensors need calibration, and that tuning will improve models. Figure 8 should be omitted. If you are insistent on including something like this, an analysis showing the statistical significance in model improvement might be more impactful.

Is there more analysis or more takeaways to be had from Table 12? All the text is really saying is that the numbers in the table match the numbers in the figure. Stronger analysis in the text is needed to make the table worth keeping.

Is there a better way to visualize the information in table 13? It's inclusion is helpful, but a figure could be more informative than a table.

Table 14 contains repetitive information and should be removed.

Table 15 would be more useful if combined with table 1 instead of expecting the reader to remember the sensor specs from the very beginning. However, as the authors point out, this is comparing multiple different types of sensors that aren't inherently comparable. I understand that the authors are trying to show the usefulness of their calibration, but I don't think they need to directly compare with others for that message to come across. I recommend removing tables 1 and 15, especially because the inclusion of information on these other sensors in the earlier sections muddles the message of what the paper is ultimately trying to convey.

Line 321: This paragraph isn't indented, but all the others in this section are.

Line 324-325: Which model are these statistics from? The abstract suggests GB, but this should be clearly stated in the conclusions as well.

Line 350: Missing a period at the end of the sentence.

---

## Author Response (AR2)

**REFERENCE: amt-2024-127- "Reviewer 1" Round 2**

**Title: "***Improving Raw Readings from Low-Cost Ozone Sensors Using Artificial Intelligence for Air Quality Monitoring***"**

**Authors:** *Guillem Montalban-Faet, Eric Meneses-Albala, Santiago Felici-Castell, Juan J. Perez-Solano and Jaume Segura-Garcia*

Departament de Informàtica, ETSE, Universitat de València, Avd. de la Universidad S/N, 46100 Burjassot, (Valencia), Spain

Dear editor and reviewer,

Thank you for giving us the opportunity to address the comments provided by the anonymous reviewers. We have made every effort to respond thoroughly to their feedback. Attached is a response letter with our responses highlighted **in magenta for this Round 2**. The revised manuscript also uses magenta text to indicate the changes made, keeping the changes from **Round 1 in blue**.

We would also like to express our gratitude to the anonymous reviewers for their valuable comments and suggestions. We appreciate the time and effort they have invested in improving our work. We firmly believe that this manuscript is now suitable for publication and an excellent contribution to share with the broader research community.

**Reviewer's comments (Referee #1 Round 2, March 10 2025) Round 2**

**First of all, we would like to sincerely thank you for your thoughtful review and comments, which have greatly contributed to improving our work.**

**In the following sections, we will address all your comments, queries, and suggestions.**

Summary: While this draft shows improvement, **more work is needed on the introduction/related work** to set the stage for a strong paper. These sections should clearly set up: 1) what is **already** being **done** in the space; 2) what is **lacking** in the space; 3) what you will do differently to **expand** on what's already been done. There are plenty of other papers already using ML, GB., etc. – what about your model is different?

Likewise, **figures and tables** should be included selectively – many are still **superfluous** and either demonstrate the same information as each other, or information that is already well- established in the field. These figures should be combined or removed as appropriate.

**Response 1: Thank** you for this comment. We think that the goal and **contribution** of this draft is relatively clear, that is the accuracy improvement of **ground-level ozone** measurements from low-cost sensors but using less expensive air quality monitoring modules, in particular the ZPHS01B multisensor module. The related work and the selected papers used for comparison are using low-cost sensors **ten times more expensive** as it is detailed in the manuscript.

Moreover, since Machine Learning-based algorithms show the best results as discussed in Section 2 in the context of low-cost air quality sensors, in particular for ground-level ozone, **we have focused exclusively on them**, evaluating up to four different models, whereas other studies have only considered one or two. We follow a clear exploratory data analysis, focused on FIA, FS and a detailed HPO process for the different models. Notice that Machine Learning algorithms are the ones that best adapt to the nonlinearities of these sensors, compared to statistical approaches.

In addition, in our models, we include the "**date**" feature (variable), as metadata, as depicted in Section 3.3, which takes into account the effects of aging and detects additional information from road traffic patterns.

Thus, regarding the "**what**" questions:

1) what is already being done in the space; *There are many contributions, and to the best of our knowledge all of them considered in Section 2.*

2) what is lacking in the space; *There is always room for improvement through different aspects: different models and their design, exploratory data analysis, better and different features (variables) and new sensors and platforms to name a few.*

3) what you will do differently to expand on what's already been done: *In our case, we achieve similar or better results with cheaper sensors (10 time less expensive) in an environment with lower ozone concentrations (with a mean value of 55.72 ug/m$^3$), including all the sensors from ZPHS01B (9 in total) and metadata ("date") in the machine learning process, in a well-defined structured approach for exploratory data analysis. The metadata is used to account for the aging effect and improve the models following road traffic patterns.*

Notice that in addition to our previous manuscript, **we have reviewed and updated the state of the art across various bibliographic databases** from the most important publishers. In particular, we searched for journal publications in IEEE (excluding IEEE Access), Elsevier (ScienceDirect) and Copernicus. Our search focused on calibration methods for low-cost ozone sensors using ML techniques. In practice, there are not that many publications on this topic. Narrowing the search by subject, we found around 50 publications, and after reviewing their contents, we identified only **3 recent references** with a truly similar focus and could be added to update the list of references already included. Briefly, the discarded publications were excluded because they either dealt with tropospheric ozone, integrated additional satellite imaging systems, focused on prediction rather than calibration, used ML for other air quality parameters (without including ground-level ozone), or focused specifically on deep learning (DL).

These new references are:

Cavaliere, A., Brilli, L., Andreini, B. P., Carotenuto, F., Gioli, B., Giordano, T., Stefanelli, M., Vagnoli, C., Zaldei, A., and Gualtieri, G.: Development of low-cost air quality stations for next-generation monitoring networks: calibration and validation of NO₂ and O₃ sensors, Atmospheric Measurement Techniques, 16, 4723–4740, https://doi.org/10.5194/amt-16-4723-2023, 2023.

Wang, G., Yu, C., Guo, K., Guo, H., and Wang, Y.: Research of low-cost air quality monitoring models with different machine learning algorithms, Atmospheric Measurement Techniques, 17, 181–196, https://doi.org/10.5194/amt-17-181-2024, 2024.

Wang, R., Li, Q., Yu, H., Chen, Z., Zhang, Y., Zhang, L., Cui, H., and Zhang, K.: A Category-Based Calibration Approach With Fault Tolerance for Air Monitoring Sensors, IEEE Sensors Journal, 20, 10756–10765, https://doi.org/10.1109/JSEN.2020.2994645, 2020.

These 3 new references included have been explained and included in Section 2 as follows:

The calibration process of these LCS is a challenge as mentioned before, where ML and Deep Learning (DL) models can be used. In (Wang et al. (2024)), a low-cost multi parameter AQ system based on $PM_{2.5}$, $PM_{10}$, $SO_2$, $NO_2$, $CO$ and $O_3$, along with Temp and RH is proposed using and evaluating various calibration algorithms. For $O_3$, the algorithms are ranked from best to worst fit as follows: RF, K-Nearest Neighbors (KNN), Back Propagation (BP), Genetic Algorithm Back Propagation (GA-BP), and Multiple Linear Regression (MLR), with $R^2$ values (MAE, in $\mu g/m^3$) of 0.98 (2.88), 0.87 (7.33), 0.83 (11.14), 0.83 (10.90), and 0.74 (13.46), respectively. With a mean $O_3$ concentration of approximately 70 $\mu g/m^3$, as shown in their Figure. 12, the RF model achieves a MRE of 4.11%. In (Cavaliere et al. (2023)), based on $O_3$ and $NO_2$ metal oxide sensors, along with Temp and RH, the authors analyzed different calibration options using uni-variate/multi-variate, linear/non-linear and parametric/non-parametric approaches with algorithms such as Linear Regression (LR), Non-Linear Regression (NLR), Support Vector Machines (SVM), RF and GB. They concluded that Multiple Random Forest (MRF) achieved the highest accuracy during Phase I (pre-deployment), with an $R^2$ of 0.98 and MAE (MRE) of 4.31 (5.74%), considering a mean $O_3$ during their deployments of 75 $\mu g/m^3$, depicted in their Figure 3 and 7. However, in Phase II (field validation) conducted at a different location, the performance worsened, with the MAE (MRE) 22.22 (29.62%) while MLR 12.96 (17.28%). In this case, MLR provided better results. The authors conclude that MLR may be a more suitable solution for representing physical models beyond the Phase I calibration dataset, demonstrating better transferability across diverse spatial and temporal settings, highlighting that parametric models such as MLR have a defined equation with only a few parameters, making them easier to adjust for potential changes over time. In (Wang et al. (2020)), the authors propose a category-based calibration approach (piecewise) using ML, which builds separate regression models for different pollutant concentration levels. This proposal is tested on $CO$ and $O_3$ data from two Chinese cities, Fuzhou and Lanzhou, with good and bad AQ, with mean $O_3$ concentrations of 69.545 $\mu g/m3$ and 49.781 $\mu g/m3$ respectively for 11 months (48 weeks). The achieved metrics for the best results are given by Extreme GB and Light GB machine algorithms (outperforming linear regression and RF) with MAE ($\mu g/m3$) of 10.75 and 10.98 in Lanzhou city respectively, and 13.83 and 14.98 in Fuzhou city, with a MRE greater than 19.88%.

And in Table 11 (in the new version), we have included also these 3 references (the first 3 row) for comparison as follows:

| Study | Location | Platform, Sensor | $R^2$ | MRE [%] | Comment |
|---|---|---|---|---|---|
| (Wang et al. (2024)) | Zhengzhou (China) | by Hanwei Electronics Corp, $O_3$ B4 Alphasense | 0.93 | 4.11 | 52 weeks dataset with RF and HPO |
| (Cavaliere et al. (2023)) | Florence, Montale (Italy) | AirQino LC, $NO_2$ MiCS-2714, $O_3$ MiCS-2614 | 0.98 with MRF | I:MRF 5.74. II:MLR 17.28, MRF 29.62 | 61 weeks dataset with MRF and MLR, using complete EDA |
| (Wang et al. (2020)) | Lanzhou (China) | Sailhero instrument, - | - | 19.88 | 48 weeks dataset, category-based calibration (piecewise) with Extreme GB and FS |
| (Zimmerman et al. (2018)) | Pittsburg (USA) | RAMP, Alphasense Ox-B431 | 0.86 | 15 | 16 weeks dataset with RF |
| (Esposito et al. (2016)) | Cambridge (UK) | SnaQ, Alphasense B4 Electrochemical | 0.69 | 42 | 5 weeks dataset using Dynamic NN with a kind of HPO |
| **Our model** | Valencia (Spain) | ZPHS01B, Winsen ZE27 | 0.93 | 7.21 | 57 weeks dataset using GB with FIA, FS and HPO |

Table 11. Comparison with similar related works

We agree that the Machine Learning techniques used are not innovative, but their use as a tool to achieve the accuracy shown in our results, combined with the methodology, following the recommendations and best practices reported in other scientific works and dealt in the manuscript, make it relevant and interesting for the community (i.e. for researcher and practitioners). These steps and methodology, as well as their explanation and justification are not found in the related work as is highlighted in the new version of the manuscript, as stressed in the previous revision.

We emphasize that some of the steps for data preprocessing, analysis and interpretability are often overlooked, such as Feature Importance Analysis (FIA), Principal Component Analysis (PCA) and Feature Selection (FS). In this line, in the manuscript it is said that the process of optimizing algorithms through the selection of their hyperparameters is also neglected in some environmental research studies. As we mentioned before, all these details are already included in Section 2 "Related work" and checked with these representative papers.

It is worth mentioning that we are combining two different disciplines, air quality and artificial intelligence. And it is difficult to master both disciplines and this is the reason in Section 2 we go into detail with these aspects, checking if the procedures used in the related works are overlooking these steps in these papers.

Thus, **in summary our work provides several contributions of interest,** which we list below:

1. **The multisensor module ZPHS01B is priced at approximately 150 euros and includes 9 different sensors**, making it cheaper than other systems used in the state of the art. This is a distinguishing starting point and of interest to the scientific community. In the related work, systems considered low-cost typically refer to a price of less than 150 euros **per sensor**.

2. **The approach of using a single module with all 9 integrated air quality sensors**, which **enables the evaluation of cross-sensitivity issues** between sensors and their potential added value, is another differentiating element. In the state-of-the-art works reviewed, the systems generally use separate sensors of different types and features, which can be interchanged. We have as many different ozone sensors as there are papers on the related work.

3. **In the studies compared in Table 11 of the new version of the manuscript**, various methods are used for calibration, **including statistical methods (also known as white-boxes) and machine learning (ML)-based approaches (grey-boxes)**, the latter of which tend to yield better results. However, in those comparisons with **ML, only one or two methods are usually applied.** In our case, **we focus exclusively on machine learning and perform a more in-depth evaluation of four different methods**.

4. **Due to the design and characteristics of these low-cost sensors, aging affects their performance over time.** Only in the mentioned article by (Cavaliere et

al.(2023)), it is proposed an adjustment for a linear regression method to account for this, but the adjustment is not applied to other methods, especially not to those based on machine learning. In our machine learning models, **this effect is incorporated through the "date" feature**, which—while technically a **metadata** field—helps reduce error in the models by not relying solely on environmental variables. This feature allows us to capture both the effect of sensor aging and pollution patterns associated with traffic from combustion vehicles. Besides, this feature (date) will allow us to improve the models following the road traffic patterns.

5.   **We already evaluated Deep Learning (DL) methods at the request of Reviewer 2 in Round 1 of the revision process** (as can be seen in the discussion forum of the platform), and we also extended the initial dataset from 165 to 239 days. Although DL results are not directly included in the article, we provide a relevant discussion explaining why such methods were discarded in this scenario, as noted in section 3.4. In fact, **we observed that for the datasets obtained during the measurement campaigns, DL models tend to learn and memorize the dataset entirely, leading to overfitting.** We believe this is due to the intrinsic characteristics of the air quality monitoring scenario and the behavior of the low-cost sensors in the ZPHS01B module, **as the datasets generated are limited and constrained for the use of DL techniques.** This is the reason we do not include these results. These results are shown in the response to this Reviewer 2 with $R^2$=0.9999999999976741, RMSE: 3.514481801502949e-05 and MAE: 2.9925663790820442e-05.

Besides, notice that Machine Learning (ML) models are often more practical, efficient, and interpretable for time-series prediction tasks, especially when datasets are small-to-medium-sized, contain noise, or require explicit domain knowledge. While DL models like LSTMs and TCNs excel in capturing long-term dependencies in very large datasets. Thus, with our dataset, we observed that DL techniques are not able to generalize as the ML approach did. And for this reason, the results using DL techniques are not so robust and reliable, mainly due to overfitting.

**6.** The results presented share the same characteristics as those presented in **Table 11 of the new version of the manuscript** used for comparison. However, there are two important differentiating elements. On the one hand, as mentioned before, the module used is significantly more affordable (around 10 times cheaper). On the other hand, **the results reported in the referenced works (Wang et al. (2024)) were obtained in environments with much higher ozone concentrations**. It is important to note that these sensors perform worse at **low concentrations** than at high ones due to their sensitivity limitations and the weakness of the signals generated, as well as **interference from other pollutants**. While in our dataset the ozone concentration is lower, with an average ozone concentration of **55.72 μg/m³**—i.e., in the cited studies the values are higher (more than 70 μg/m³). This information is included in the new version of the manuscript as follows:

not all of these works follow and discuss an structured EDA with FIA, FS and HPO. **In particular, when compared to the first two works with slightly better results, in (Wang et al. (2024), we appreciate higher $O_3$ values, mean values higher than $70\,\mu g/m^3$, while in our case we have lower levels ($55.72\,\mu g/m^3$)), as well as there is not a complete EDA. It is important to note that these sensors perform worse at low concentrations than at high ones due to their sensitivity limitations and the weakness of the signals generated, as well as interference from other pollutants. Finally, in (Cavaliere et al. (2023)), although the authors use a complete EDA, they only use two sensors ($NO_2$ and $O_3$) apart from Temp and RH, and the**

**7.** Finally, although there are a couple of articles that follow a more structured approach, in particular (Cavaliere et al.(2023)), **most do not carry out the recommended steps required to properly apply machine learning algorithms**, such as conducting exploratory data analysis and including **Feature Importance Analysis (FIA), Feature Selection (FS) and Hyperparameter Optimization (HPO) stages.**

All these comments have been incorporated into the wording of the new version, particularly in the abstract, at the end of Section 2 (Related Work), and in the conclusion.

Finally, about the figures and tables included in the manuscript, they are discussed in the following responses.

We have clarified this issue in the new version of the manuscript.

**Subscripts** are needed throughout for $O_3$, $CO_2$, etc.

Abstract: Readers will know what ozone is. This space would be better spent explaining why **you need machine learning enabled calibration**.

**Response 2:** Thank you for your comment. We have updated the subscripts for the chemical formulation.

In the abstract, regarding the calibration process in general, it arises from a lack of accuracy in low-cost ozone sensors. However, we have improved the wording for clarity as well as a better justification of the machine learning techniques used in this case, but in a brief form for the abstract, as follows:

**Abstract.** Ground-level ozone ($O_3$) is a highly oxidizing gas with very reactive properties, harmful at high levels, and generated by complex photochemical reactions when primary pollutants from the combustion of fossil materials react with sunlight. Thus, its concentration indicates the activity of other air pollutants and plays a crucial role in smart cities. With the growing interest in high-resolution Air Quality (AQ) monitoring, low-cost ozone sensors present an interesting alternative, although they lack accuracy and suffer from cross-sensitivity issues. In this context, artificial intelligence techniques, particularly ensemble Machine Learning (ML) models, can improve the raw readings from these sensors by incorporating additional environmental information to minimize inaccuracies and nonlinearities, as well as by including metadata to account for sensor aging effects and improve the models based on road traffic patterns. In this paper, based on the low-cost ZPHS01B multisensor module with nine sensors, we analyze, propose, and compare different techniques using four ML models in a low $O_3$ concentration scenario (mean value of 55.72 $\mu g/m^3$). We carried out a thorough exploratory data analysis process to extract the main features (variables) and performed hyperparameter optimization for the different models. As a result, we reduced the estimation error by approximately 94.05%. In particular, using the Gradient Boosting algorithm, we achieved a Mean Absolute Error (MAE) of 4.022 $\mu g/m^3$ and a Mean Relative Error (MRE) of 7.21%, outperforming related work while using a module approximately ten times less expensive. To carry out this work, we generated two datasets in the city of Valencia (Spain), at two different locations with the same characteristics (close to the ring road but separated by 4.1 km), with 165 and 239 days.

Line 14: do the authors ever come back to these **guidelines**? If not, this paragraph is not useful. Same with the next paragraph – these standards are not really mentioned again later. I understand that this is trying to establish the "why use low-cost sensors", but it needs to be more clearly related back to what you're actually doing.

**Response 3:** The reviewer is correct. In Section 1, we introduce, motivate, and contextualize the problem of ground-level ozone based on the Air Quality Guidelines and the objectives set within the mentioned directives. This approach is twofold: on one hand, we focus on air pollutants (particularly ozone) and their impact on health, and on the other hand, we emphasize the importance of higher spatial monitoring resolution for these pollutants.

However, we have specifically improved the wording in the Conclusion section to address this issue and revisit the problem statement covered in this manuscript, as introduced in Section 1. Thus, in the Conclusion, we refer to these standards and guidelines again for closure.

**5 Conclusions**

365 This paper focuses on ground-level ozone ($O_3$), as it serves as an indicator of other pollution levels in urban areas using LCS nodes based on the ZPHS01B module. These nodes will enable an increase in the spatial sampling of AQ monitoring in cities, following the interest of AQG (Organization et al. (2021)) and in line with the future plans of the related directives, ideally at least one sample per 100 $m^2$, according to Annex III-B of the European (Directive 2008/50/EC (2008))..

Line 27: Why is "**primary**" in quotes but not secondary? Be consistent, but quotes are not necessary. There are also quotes around primary on line 2.

Line 34: What is "**official** equipment"?

Line 51: While not detrimental, this paragraph is unnecessary.

**Response 4:** Thanks for these corrections. We have removed the "quotes" for primary and secondary. About "official equipment" expression, maybe the term official is not

adequate, and it should be better "regulated", "certified" or "standardized". We have explained this and changed this expression. Thus, we refer to regulated equipment, when we refer to "standardized air quality monitoring stations".

The paragraph in line 51, although it is often found in the research papers, to assist the reader, we have omitted it. If the editor considers incorporating it, it has been just commented % in latex in the source files.

We have revised this expression in the manuscript accordingly.

Related works: see comments about **table 15**, but the information on the specific other **sensors** used for comparison could be restructured, if not removed. The information added here on specific ML **models here is helpful, but it could be improved further by exploring more clearly the strengths and weaknesses of each of these**, and how you will improve upon this and not just repeat what's already been done.

**Response 5:** Thanks for this comment. Regarding this table, now **Table 11 of the new version of the manuscript**, we have included some extra details for further information for clarity. Notice that this table only considers some of the modules shown in Table 1, in particular RAMP, AirSensrEUR and ZPHS10B.

Moreover, we highlight that the goal of Table 1 is to compare different commercially available low-cost multisensor modules and alternatives, detailing only their sensors and price range, without considering whether all these modules have been used in related work.

Finally, about the strengths and weaknesses of the related work, it was already considered in the previous review, in blue as follows:

good example of the use of these good practices is shown in (Cavaliere et al. (2023)). In addition, in (Zhu et al. (2023)),

145 it is said that the process of optimizing algorithms through the selection of their hyperparameters (Hyperparameter Optimization (HPO)) is neglected in most of the environmental research studies considered. For instance, in (Johnson et al. (2018)), better results are obtained with GB, but HPO is not performed in the model, which could allow further improvements of the results. In (Malings et al. (2019)), (Wang et al. (2020)) and (Zimmerman et al. (2018)), it is taken into account some aspects related to the data analysis focused on the optimization of the problem, but they do not carry

150 out a HPO. In (Esposito et al. (2016)), the authors carry out a kind of simple HPO, based on raw tests of different architectures and modifying hyperparameters, such as the number of hidden layers of the model, tapped delay length and feedback delay line length, concluding that a dynamic approach to these parameters improves the results with respect to a static approach without changing the value of these parameters.

Regarding the selection of parameters, in (Johnson et al. (2018)), the authors does not perform an analysis using tech-

155 niques such as the aforementioned FIA and FS, but a sensitivity analysis using different meteorological variables (such as Temp and RH), determining that it is useful information for GB. In (Malings et al. (2019)), the quantification of the importance of the model variables is mentioned as a mean to understand which information is useful, concluding that for RF, to add additional information apart from AQ measurements, such as Temp and RH are very helpful. In (Esposito et al. (2016)) and (Wang et al. (2024)), the authors do not include a specific analysis of the relative importance of

160 different variables or features. However, a good example of FS is depicted in (Okafor et al. (2020)), where it is shown that identifying the environmental factors affecting LCS is crucial for improving data quality using data fusion and ML. These factors are then incorporated into the development of the calibration model.

In conclusion, in order to increase the resolution of city-scale AQ monitoring according to the recommendations given by (Directive 2008/50/EC (2008)) as mentioned before, it is necessary to perform a calibration process of these LCS. In

165 this scenario, we focus on $O_3$ calibration using ensemble ML techniques to minimize inaccuracies and nonlinearities, comparing four different models, considering different environmental variables as well as metadata mainly to account for sensor aging effects. For this purpose, it is necessary to carry out a thorough data treatment with a good practice criteria (Zhu et al. (2023)) including HPO, FIA and/or FS, which are usually overlooked. In a scenario with low $O_3$ concentration, we achieve interesting results compared with the related work, as shown in Section 4.

**Table 3: This table should be removed.** There are still no units in this table (temp, RH, PM2.5, CO2, NO2, CO, etc. should all have units attached). The statistics of the measured quantities are not referenced or used anywhere else in the paper, and the reader can't do anything with this information on their own. **Likewise, "stationarity" and "percentage of samples taking Different values" are not analyzed further in the text**. The paragraph beginning on line 181 can be condensed to give the context the table is hoping to provide (**ex. "Sensors X, Y, and Z appeared particularly unreliable and were omitted from our model".**)

**Response 6:** Thanks for this comment. We have introduced these units both in the caption and in the text.

However, regarding the content of this table, note that the ZPHS01B model has not been previously used for these issues. For this reason, it is important for us to justify our choice and provide all relevant details and evidence to clarify and characterize its behavior.

The statistical analysis conducted with the datasets may seem redundant if using the same module as other research groups or well-known sensors, but this is not our case. Nonetheless, **we have simplified this table** by removing ' Variance (Var.), Stationarity (Stat.) and 'Seasonality (Seas.)', retaining the more relevant statistics.

In addition, these statistics are used in the results section to calculate additional metrics and parameters, in particular when we estimate the mean relative error.

The new version of this table 3 is as follows:

Table 3. Summary of main statistics of the Dataset: Minimum (Min.), Maximum (Max.), Mean (Mean), Standard Deviation, Median Absolute Deviation (MAD), percentage of samples taking Different values (Diff.) and High correlation (High corr.)

| | Temp [ºC] | RH [%] | $PM_{2.5}$ $[\mu g/m^3]$ | $CO_2$ [ppm] | $NO_2$ $[mg/m^3]$ | CO $[mg/m^3]$ | $CH_2O$ $[mg/m^3]$ | TVOC [levels] | $O_3$ $[\mu g/m^3]$ | $O_3ref$ $[\mu g/m^3]$ |
|---|---|---|---|---|---|---|---|---|---|---|
| Min | 5.24 | 62.29 | 21.25 | 693.43 | 0.78 | 0 | 0.005 | 0 | 39.57 | 8.71 |
| Max | 42.26 | 118 | 83.69 | 1792.50 | 18.81 | 0.75 | 1.21 | 2.95 | 255.76 | 97.85 |
| Mean | 20.60 | 91.31 | 49.99 | 780.33 | 15.27 | 0.34 | 0.021 | 0.024 | 114.39 | 55.72 |
| SD | 5.70 | 18.12 | 18.14 | 57.16 | 5.65 | 0.28 | 0.02 | 0.13 | 67.11 | 24.83 |
| MAD | 3.92 | 16.37 | 13.31 | 24.53 | 0.59 | 0 | 0.001 | 0 | 51.40 | 16.21 |
| Diff. | 99.1% | 81.9% | 87.9% | 97.5% | 50.6% | 0.2% | 81.2% | 5.8% | 75.0% | 30.3% |
| High corr. | yes | yes | yes | yes | not | yes | not | not | yes | yes |

Figure 3: Any ambient pollutant will have a **repeating diurnal pattern** from the boundary layer rising and falling each day, and most sensors will pick up on major sources like traffic.

A DFT is not necessarily needed to show this and confuses the messaging in this section. Since this figure is never referenced again other than to show that a pattern exists, it should be omitted.

**Response 7:** Of course, a repeating diurnal pattern associated with the day/night cycle is evident once we analyze the DFT, as it reveals ground-level ozone generation through photochemical reactions.

However, when the selected sensors are under test (especially in a low-cost approach like this) and this analysis has not been previously performed, we do not consider this check redundant. We believe that we should not assume certain patterns as obvious without verification.

It is possible that this pattern does not exist or cannot be detected with these sensors, which is precisely why we applied this analysis. In our dataset, particularly for ozone, the pattern is clearly observable, and this method provides a straightforward way to demonstrate it.

Perhaps this analysis has not been included in previous related work because researchers have used well-known low-cost sensors.

That said, we are open to removing this information if the editor deems it unnecessary.

Meanwhile, **we have placed this information in "Appendix A:** Spectral analysis for O3 low-cost readings from ZPHS01B module" of the new version of the manuscript.

**Figure 4:** While this figure is fine, it's well known in the low-cost sensor space that sensors can capture the general trends of pollutants but need calibration to accurately convey the magnitude. This figure should be **omitted**.

**Response 8:** As we stated previously, the ZPHS01B module has not been used before in this kind of studies and research. Note that we selected this module for several reasons, as explained in the manuscript. It offers the best price-per-sensor and price-to-quality ratio, embedding 9 different sensors on the same board.

Thus, the data provided by these sensors is valuable for analyzing cross-sensitivity issues, enabling the training of different calibration models and extracting more information than would be possible with single sensors.

For this reason, examining and demonstrating the behavior of the $O_3$ sensor in this module is particularly relevant. For instance, we observe a positive offset in the raw readings compared to the regulated and standardized $O_3$ measurements from the AQ station. This trend was also reflected in the error distribution shown in Figure 8, that finally was removed in the new version of the manuscript as it is suggested later in Response 15.

**Table 4**: If this is all to make a better ozone model, the **FIA** of ozone should be included here to show how much it improves the model. How was **8% importance** selected? It sounds arbitrary. It would also be easier on the reader if the threshold and the table were in the same format (either **both in decimal or both in percent)**.

**Response 9:** Thank you for your comment. Table 4 presents the normalized output of the FIA using the scikit-learn library for parameters complementary to ozone, for each model used. For clarity, all contributions are expressed per unit (1). From this table, we observe the following:

- On one hand, **Temp, RH, and $CO_2$** exhibit higher contributions compared to the other parameters. We have highlighted these values in bold.

- On the other hand, **$NO_2$, $PM_{2.5}$, $CH_2O$, TVOC, and CO** show lower contributions, falling below the suggested heuristic threshold of 0.08 (8%), as no other criterion applies in this case. Additionally, $NO_2$, $CH_2O$, TVOC, and CO were already discussed in the analysis of Table 3, except for $PM_{2.5}$.

We have **refined the wording and improved this table** in the manuscript regarding its analysis as follows:

**Table 4.** FIA of ozone's complementary parameters **for Random Forest (RF), Gradient Boosting (GB), Adaptive Boost (ADA) and Decision Tree (DT)**, in bold the selected ones, contribution higher than 0.8.

| Model | Temp | RH | PM$_{2.5}$ | CO$_2$ | NO$_2$ | O$_3$ref | CO | TVOC | CH$_2$O |
|-------|------|------|------|------|------|------|------|------|------|
| **RF** | **0.128** | **0.103** | 0.069 | **0.222** | 0.078 | **0.269** | 0.002 | 0.003 | 0.064 |
| **GB** | **0.107** | **0.105** | 0.052 | **0.211** | 0.057 | **0.253** | 0.001 | 0.001 | 0.068 |
| **ADA** | **0.119** | **0.097** | 0.064 | **0.246** | 0.067 | **0.287** | 0.001 | 0.001 | 0.066 |
| **DT** | **0.115** | **0.088** | 0.070 | **0.232** | 0.061 | **0.276** | 0.001 | 0.002 | 0.061 |

230    Table 4 shows the normalized output of the FIA using the *scikit-learn* library (Pedregosa et al. (2011)), for the parameters complementary to $O_3$, for each ML models used. **In order to determine the most useful parameters for the models, a threshold is established in 0.08, that is 8% of importance. These parameters are in bold. Notice that the set of parameters with the highest importance, is repeated for all models: Temp, RH, $CO_2$ and $O_3$.**

Besides, we must highlight that these are preliminary steps, and it does not mean that we directly will exclude these parameters with lower contribution at this point.

**Figure 5** is essentially showing that some sensors are more cross sensitive than others, which is already well established in the field. This figure should be **omitted**.

**Response 10:** In our opinion, this figure could be considered redundant when dealing with well-known and well-characterized sensors. However, this is not our case. In line with the arguments mentioned in Response 8, the ZPHS01B module has not been used in a similar way before. Thus, the **information provided is valuable for analyzing cross-sensitivity issues. This type of information is part of the exploratory data analysis (EDA) and feature selection (FS).**

**Tables 5, 6, 7 and 8 should be combined into a single table with the 4 sub-categories as another column**.

**Response 11:** Since we are dealing with four different models, each with different hyperparameters (both in number and meaning), it is clearer to present this information in separate tables. These tables are different and **cannot be combined** in a clear way.

**Table 9** is unnecessary and can be omitted – you and many others have already established that **hyperparameter tuning will make the models fit better**.

**Response 12:** Thanks. We have **omitted** this table and left only the optimized versions. Simply, we have just introduced a sentence detailing how much improvement the hyperparameter optimization introduces in the different models, as follows:

320    It is worth mentioning that the improvement achieved by HPO is greater in GB and ADA models than in RF and DT, which are already well-optimized with default values. In particular, for the optimized GB and ADA models, $R^2$ is improved by 42% and 182%, respectively, while RMSE is reduced by 57% and 66%. However, the execution time required for training is influenced by HPO, increasing to 66.937s and 7.805s for GB and ADA, respectively, as shown in Table 9. We highlight that RF and DT are already well-optimized, and their execution times remain unchanged between

325    the default and optimized versions.

Tables **10 & 11** should also be **combined**.

For tables 9, 10, and 11, and Figure 6, it is not specified in the titles whether it is **training or testing data** – please specify.

In the low-cost sensor field, **it is standard to show both training and testing data - consider adding to tables 9, 10, and 11, and Figure 6.**

**Response 13:** Thanks for your comment. We have combined both tables in one as follows:

**Table 9.** Performance metrics for HPO models with 90/10 and 80/20 (training/testing) ratio

| Model | GB | | RF | | ADA | | DT | |
|---|---|---|---|---|---|---|---|---|
| Ratio | 90/10 | 80/20 | 90/10 | 80/20 | 90/10 | 80/20 | 90/10 | 80/20 |
| R² | 0.938 | 0.936 | 0.927 | 0.924 | 0.922 | 0.920 | 0.878 | 0.863 |
| RMSE | 6.492 | 6.664 | 7.093 | 7.253 | 7.289 | 7.416 | 9.149 | 9.735 |
| MAE | 4.022 | 4.221 | 4.185 | 4.415 | 3.642 | 3.833 | 4.684 | 5.104 |
| MAPE | 0.194 | 0.206 | 0.208 | 0.228 | 0.160 | 0.175 | 0.206 | 0.226 |
| Time | 66.937 | 61.054 | 18.316 | 16.618 | 7.805 | 7.078 | 0.212 | 0.194 |

The results shown are **always for testing data** as it is detailed in the manuscript. We do not show the training process. However, in the next response 14 we will discuss this issue again, proving the results from training and validation.

Notice that we split the dataset for training and testing, both sets remain independent and isolated with different training-test ratio percentages: 60%-40%, 70%-30%, 80%-20% and 90%-10%.

And during the training process itself, the dataset is further divided into two parts: one for training and the other for validation. By default, we allocate 80% of the data for training and 20% for validation. In this process, the training and validation datasets are combined across different iterations.

We have improved the wording to clarify this issue in the new version of the manuscript as follows:

295  We evaluated the performance metrics of these ML models under different configurations (in terms of $R^2$, RMSE, MAE in $\mu g/m^3$ and Mean Absolute Percentage Error (MAPE) and execution time in seconds), with the optimized hyperparameters that achieve higher $R^2$ and lower errors. Also, we used the three different datasets given by different monitoring intervals: 10 and 30 min and 1 h, as depicted in Section 3.2. We tested different training-test ratio percentages from these datasets: 60%-40%, 70%-30%, 80%-20% and 90%-10%, denoted as 60/40, 70/30, 80/20 and 90/10. Note that when we split the dataset

300  for training and testing, both sets remain independent and isolated. However, during the training process itself, the dataset is further divided into two parts: one for training and the other for validation. By default, we allocate 80% of the data for training and 20% for validation. In this process, the training and validation datasets are combined across different iterations. From all of them, we have achieved the best results in terms of these performance metrics with

Is the point of Figure 7 just to show that the model isn't **overfitting**? It needs more analysis in the text rather than relying on the reader to interpret.

**Response 14:** As mentioned above, in Response 13, **we only show the results from testing data.**

However, in the training process, the used dataset (excluding testing dataset) is further divided into two parts: one for training and the other for validation, by default 80% for training and 20% for validation respectively during the different iterations.

Since the convergence of performance metrics provides information about overfitting for both the training and validation datasets, we have included the following plots, which show the R² and RMSE values across different iterations during the training process for various models. Each model uses a reference hyperparameter for convergence.

[Figure]

We can observe in the above plots during the different iterations the fit of the model in terms of R², with a better fit with training than with validation, as expected. In addition, it should be noted that the convergence process with training does not reach a perfect fit in any case, which justifies and supports the conclusion that there is no overfitting in the models.

Moreover, we see that the achieved R² score for both training and validation is better than the values shown for testing, which are the ones included in the tables in the manuscript. That is because the testing dataset does not participate in the training process.

[Figure]

About RMSE, the above plots show a similar behavior during different iterations, with a better fit in training compared to validation. As mentioned before, these values are better than those shown in the table from the testing process.

All this information, figures and explanation, has been included in "Appendix B: Results of models' convergence".

Again, it is well-established that low-cost sensors need calibration, and that tuning will improve models. **Figure 8 should be omitted**. **If you are insistent on including something like this, an analysis showing the statistical significance in model improvement might be more impactful.**

Is there more analysis or more takeaways to be had from **Table 12**? All the **text** is really saying is that the numbers in the table match the numbers in the figure. Stronger analysis in the text is needed to make the table worth keeping.

**Response 15:** Thanks. We have omitted Figure 8 and Table 12, which illustrates the error distribution and its analysis, related to standard deviation and confidence intervals.

We found it interesting to observe how the adjustment in the calibration process is carried out based on the raw readings, allowing us to identify default deviations and tendencies directly from the embedded sensors in the ZPHS01B module. Since this module has not been used before, as previously mentioned, this information could shed light on important insights into its performance.

Besides, a statistical study based directly on this distribution is more robust and comprehensive. For instance, we could observe how the offset shown in the raw readings in Figure 4 appears as an asymmetry (skewness) in the error distribution.

However, we reconsidered and concluded that the information provided could be omitted, and it was removed.

Is there a better way to visualize the information in **table 13**? It's inclusion is helpful, but a **figure** could be more informative than a table.

**Response 16:** Thanks. Table 13 summarizes the metrics provided by the Gradient Boosting model for the different datasets used for generalization. In this case, we present the performance metrics for Dataset-1 and Dataset-2 with Node 1 and Node 2, respectively. Additionally, we have included the following bar graph for easier comparison.

[Figure]

All this information has been included in the new version of the manuscript.

**Table 14** contains **repetitive** information and should be removed.

**Response 17:** Since one of the metrics used in the comparisons is the improvement relative to the raw values, we have kept this table (in the new version is number 10) but improved its explanation in the text to clarify these results as follows:

In Table 10, we show the improvement in % using the different ML models for the calibration process from the LCS raw
345   readings of the module, highlighting the better performance of GB model compared to the other models. **Notice that with this**

**model, GB, the initial MAE from the raw readings was $67.59\,\mu g/m^3$ reducing it to $4.022\,\mu g/m^3$, that is an improvement of $94.05\%$ as depicted in this table.**

**Table 15** would be more useful if combined with **table 1** instead of expecting the reader to remember the sensor specs from the very beginning. However, as the authors point out, this is comparing multiple different types of sensors that aren't inherently comparable. I understand that the authors are trying to show the usefulness of their calibration, but I don't think they need to directly compare with others for that message to come across. I recommend **removing tables 1 and 15**, especially because the inclusion of information on these other sensors in the earlier sections muddles the message of what the paper is ultimately trying to convey.

**Response 18:** As we answered in Response 5, Table 15 (in the new version is 11) is used for comparison and we have included some extra details for clarity. In this table, we compare our models for ozone calibration for low-cost sensors, against the related work with a similar approach, highlighting the location, platform and sensors used, $R^2$, mean relative error (MRE) with comments about the details of the models used and dataset duration

Notice that this table only considers some of the modules shown in Table 1, in particular RAMP, AirSensrEUR and ZPHS10B. Table 1 is an overview of different commercial sensor modules available, detailing only their sensors and price range, without considering whether all these modules have been used in related work.

This new table (Table 11) was already shown in Response 1, and its explanations have included in the new version of the manuscript as follows:

Finally, in Table 11, we compare our models for $O_3$ calibration for LCS, against the related work with a similar approach, **highlighting the location, platform (and sensors used), $R^2$, MRE along with additional comments about the detail of**
350   **the models used and dataset duration.** First, we must stress that the starting point of the selected papers is slightly different compared to ours, since these studies have used more reliable and expensive LCS, approximately ten times more expensive that the ZPHS01B module. **Moreover, since ML-based algorithms show the best results as discussed in Section 2, we have focused exclusively on them, evaluating up to four different models, whereas other studies have only considered one or**

two. Our model, in particular GB with 4 features (including "date" as metadata), as shown in Section 3.3, achieves a MRE of
355   7.21% (given by MAE $4.022\,\mu g/m^3$ with 90/10 dataset (Table 9) and the mean $O_3$ value of $55.72\,\mu g/m^3$ (Table 3)). Besides, not all of these works follow and discuss an structured EDA with FIA, FS and HPO. **In particular, when compared to the first two works with slightly better results, in (Wang et al. (2024)), we appreciate higher $O_3$ values, mean values higher than $70\,\mu g/m^3$, while in our case we have lower levels ($55.72\,\mu g/m^3$)), as well as there is not a complete EDA. It is important to note that these sensors perform worse at low concentrations than at high ones due to their sensitivity limitations and**
360   **the weakness of the signals generated, as well as interference from other pollutants. Finally, in (Cavaliere et al. (2023)), although the authors use a complete EDA, they only use two sensors ($NO_2$ and $O_3$) apart from Temp and RH, and the**

**aging effect is considered a posteriori, while this information is included in our case by date in our models, which also detects other patterns derived from road traffic.**

Line 321: This paragraph isn't indented, but all the others in this section are.

**Response 19:** It is the default AMT template.

Line **324-325**: Which **model** are these statistics from? The abstract suggests **GB**, but this should be clearly stated in the conclusions as well.

Line 350: Missing a **period** at the end of the sentence.

**Response 20:** These details were already included in the previous manuscript. After comparing the different models, we identified Gradient Boosting (GB) as the best model and have highlighted its performance metrics in both the abstract and conclusions.

Finally, we put this period.

Finally, thank you for your thoughtful review and comments which will enable us to improve this work.  We appreciate the time and effort invested in your review.